# Drug-TTA: Test-Time Adaptation for Drug Virtual Screening via Multi-task Meta-Auxiliary Learning

Ao Shen [* 1 2]   Mingzhi Yuan [* 1 2]   Yingfan Ma [1 2]   Jie Du [1 2]   Qiao Huang [1 2]   Manning Wang [1 2]

## Abstract

Virtual screening is a critical step in drug discovery, aiming at identifying potential drugs that bind to a specific protein pocket from a large database of molecules. Traditional docking methods are time-consuming, while learning-based approaches supervised by high-precision conformational or affinity labels are limited by the scarcity of training data. Recently, a paradigm of feature alignment through contrastive learning has gained widespread attention. This method does not require explicit binding affinity scores, but it suffers from the issue of overly simplistic construction of negative samples, which limits their generalization to more difficult test cases. In this paper, we propose Drug-TTA, which leverages a large number of self-supervised auxiliary tasks to adapt the model to each test instance. Specifically, we incorporate the auxiliary tasks into both the training and the inference process via meta-learning to improve the performance of the primary task of virtual screening. Additionally, we design a multi-scale feature based Auxiliary Loss Balance Module (ALBM) to balance the auxiliary tasks to improve their efficiency. Extensive experiments demonstrate that Drug-TTA achieves state-of-the-art (SOTA) performance in all five virtual screening tasks under a zero-shot setting, showing an average improvement of 9.86% in AUROC metric compared to the baseline without test-time adaptation. The code is available at https://github.com/ShenAoAO/Drug-TTA.git.

---

[*]Equal contribution [1]Digital Medical Research Center, School of Basic Medical Sciences, Fudan University, 131 Dong'an Road, 200032, Shanghai, China [2]Shanghai Key Laboratory of Medical Image Computing and Computer Assisted Intervention, Fudan University, 131 Dong'an Road, 200032, Shanghai, China. Correspondence to: Manning Wang <mnwang@fudan.edu.cn>.

*Proceedings of the $42^{nd}$ International Conference on Machine Learning*, Vancouver, Canada. PMLR 267, 2025. Copyright 2025 by the author(s).

## 1. Introduction

Virtual screening plays a crucial role in early drug discovery by enabling the rapid identification of potential drug candidates from vast small molecule libraries for further validation (Patel et al., 2021; Schneider, 2010). The number of molecules in these libraries is increasing very fast, with recent ones reaching billions of molecules (Zhou et al., 2024). As a result, there is an urgent need for fast and efficient virtual screening methods to keep pace with the growing size of molecule libraries (Sadybekov & Katritch, 2023; Shen et al., 2024a).

Traditional virtual screening methods (Halgren et al., 2004; Trott & Olson, 2010; Spitzer & Jain, 2012; Combs et al., 2013) primarily rely on molecular docking software to predict binding conformations and estimate binding affinity by calculating the binding energy. However, these methods are often computationally expensive and time-consuming, as they require sampling a large number of conformations and evaluating each through complex scoring calculations (Kitchen et al., 2004). With the advancement of deep learning(Yuan et al., 2023; Shen et al., 2024c), the majority of current methods focus on using deep models to predict docking conformations (Cai et al., 2024; Zhang et al., 2023b) and binding affinity (Zhang et al., 2023a; Kimber et al., 2021), which significantly accelerates the screening process. However, these models are typically trained on positive pairs, i.e., binding protein-molecule pairs, with available conformational or binding affinity labels (Wang et al., 2005). As a result, the scarcity of high-quality labeled data limits their generalization ability. Furthermore, the absence of negative pairs, i.e., non-binding protein-molecule pairs, makes the training data have a large disparity with the inference data, which requires the model to identify active molecules that form positive pairs with the target protein from a large number of inactive molecules. Recently, a new structure-based feature alignment paradigm (e.g., DrugCLIP (Gao et al., 2024)) of virtual screening has emerged, which is free of complex docking simulation and high-quality affinity labels. Similar to other feature alignment approaches (e.g., CLIP (Hafner et al., 2021)), this paradigm maps molecules and pockets into a shared feature space using deep neural networks and employs contrastive learning for training,

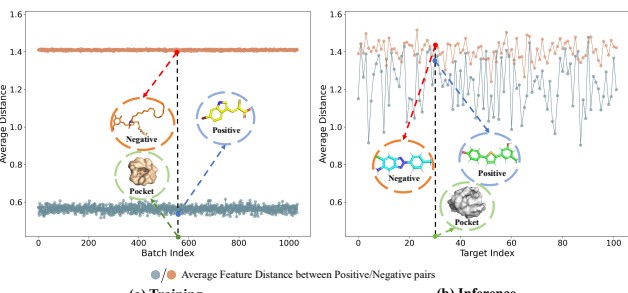

**(a) Training**      **(b) Inference**

*Figure 1.* Average feature distance between positive/negative pairs in the training and inference stages. Positive and negative pairs refer to binding and non-binding pair of protein pocket and molecule, respectively. ● and ● points represent the average feature distance for positive and negative pairs, respectively. (a) Training stage (using training dataset in DrugCLIP (Gao et al., 2024)): Each batch contains 48 positive pairs and 2,256 ($48 \times 47$) negative pairs. (b) Inference stage (using DUD-E benchmark (Mysinger et al., 2012)): There are 102 pockets and for each pocket a number of molecules are provided to form positive and negative pairs with it.

where the binding pocket-ligand data in a batch are treated as positive pairs and other pocket-ligand combinations in the same batch are regarded as negative pairs. During inference, it scores and ranks candidate molecules based on the similarity between pocket and molecule features and selects the molecules with high scores as the screening results. However, this method explores the CLIP-based approach to construct the positive and negative pairs for model training, which leads to an inherent limitation in drug virtual screening. **Specifically, negative pairs for a given pocket are defined too simplistically, consisting of only molecules binding to other pockets.** In actual virtual screening tasks, there are often challenging negative pairs, where the binding and non-binding molecules for a specific pocket exhibit very similar physicochemical properties. This results in a domain gap between the training and inference stages, making it challenging for the model to generalize effectively. To illustrate this problem, we conduct a simple statistical analysis for the training dataset in DrugCLIP (Gao et al., 2024) and the DUD-E benchmark (Mysinger et al., 2012). We calculate the average feature distance between positive pairs and negative pairs of each training batch, and the results are shown in Figure 1(a). The results reveal that distinguishing positive and negative pairs during training is relatively easy. Similarly, we calculate the average feature distance between positive pairs and negative pairs for 102 pockets in the DUD-E benchmark dataset, and the results are shown in Figure 1(b). We can see that it is much more difficult to distinguish positive and negative pairs since their feature distances are much closer than that in the training stage.

To address this issue, one naive approach could be to enhance the training dataset with more challenging negative pairs. However, designing negative pairs for training that are consistent with those encountered at test time is unreal-

istic. Given that drug screening often presents a zero-shot problem, where testing involves no labeled target-domain data, tuning training data to simulate the target domain is also not feasible. As Vladimir Vapnik (Vapnik, 2006) famously stated, "When solving a problem of interest, do not solve a more general problem as an intermediate step. Try to get the answer that you really need, not a more general one." An alternative approach is to design a network that adapts to each test sample individually. We draw inspiration from the success of test-time adaptation (TTA) in image classification tasks (Sun et al., 2020; Xiao & Snoek, 2024), which adapts a trained model to each test instance during inference to make better predictions. TTA typically leverages self-supervised auxiliary tasks during inference to dynamically update the model parameters, enabling the learning of feature representations tailored to individual test instances. This approach requires no prior knowledge of the test data distribution and instead adjusts model parameters dynamically on a per-instance basis during inference. In this paper, we propose **Drug-TTA**, which incorporates the TTA approach into DrugCLIP (Gao et al., 2024) and significantly boosts its performance on drug virtual screening. This is the first work to introduce TTA into drug screening to adapt the trained model to unseen instances at test time, and the significant performance improvement demonstrates its promise.

To leverage self-supervised learning-based TTA techniques in virtual screening, we introduce five self-supervised auxiliary tasks for proteins and small molecules, encompassing both fine-grained and coarse-grained tasks. To prevent the model from overly favoring the auxiliary tasks, which could negatively impact the performance of the primary task and result in suboptimal outcomes, we explore a meta-learning framework in Drug-TTA. During training, this framework simulates the inference process by first updating the model weights using auxiliary tasks and then applying the updated model to the primary task. This process is consistently followed throughout both training and inference, ensuring that the primary task's performance is effectively optimized by the auxiliary tasks, while mitigating overfitting to the auxiliary tasks. Furthermore, to better balance the auxiliary tasks, we design a multi-scale feature based Auxiliary Loss Balance Module (ALBM), which flexibly adjusts the weights of the auxiliary task loss during inference. This method offers greater flexibility than manually designing loss weights, and it is more effective than learning fixed weights and can generalize more effectively to new input samples during inference.

Our main contributions are summarized as follows:

- To the best of our knowledge, Drug-TTA is the first to introduce TTA into the field of drug virtual screening so that the trained model can be dynamically adapted to each test instance during inference.

- We introduce five self-supervised auxiliary tasks for TTA in drug virtual screening, spanning both fine-grained and coarse-grained tasks, enabling the model to effectively adapt to test instances. Moreover, we propose a meta-auxiliary learning strategy that allows the model to learn how to leverage these auxiliary tasks for parameter adjustment, thereby improving the performance of the primary task during inference.

- We propose a multi-scale feature based ALBM to balance the weights of self-supervised auxiliary task losses according to the test instance features, which outperforms directly assigning fixed weights.

- Drug-TTA achieves SOTA performance with significant improvements across five virtual screening tasks, showcasing its remarkable potential in drug virtual screening, particularly with an average improvement of 9.86% in the AUROC metric.

## 2. Related Work

**Drug Virtual Screening:** The goal of virtual screening is to identify the most relevant molecules from a large molecular library that have the highest probability of binding to a given protein pocket. Drug virtual screening methods can be broadly categorized into traditional methods and deep learning-based methods (Oliveira et al., 2023). Traditional methods (Halgren et al., 2004; Trott & Olson, 2010; Spitzer & Jain, 2012; Combs et al., 2013) rely on docking software to assess the binding affinity between proteins and ligands by extensively sampling and evaluating docking conformations. However, these methods are heavily dependent on the accuracy of scoring functions and require complex structural sampling and energy calculations, which reduce computational efficiency. In contrast, deep learning-based methods can significantly accelerate the screening process (Shen et al., 2024b; Du et al., 2025). Most of these methods train a deep model to predict the binding affinity between a molecule and a pocket in a supervised way with available binding affinity labels (Öztürk et al., 2018; Zheng et al., 2019; Jones et al., 2021) or calculating binding scores from known complex conformations (Cai et al., 2024; Zhang et al., 2023b). Then the trained model can be used to predict the binding affinity between a target pocket and molecules in a large library and rank the molecules to determine the best candidates. There are two issues in this kind of method. First, it is usually difficult to obtain the ground-truth affinity labels or conformation for supervised learning. Another less obvious issue is that they rely on positive pocket-ligand pairs for model training, and lack the learning of negative pairs. This makes them less aligned with most real virtual screening tasks, where the objective is to select molecules that can bind to a given pocket from a large pool of candidate molecules, most of which

form negative pairs with the given pocket. A recent feature alignment paradigm (Gao et al., 2024) avoids the need for ground-truth affinity value by projecting the pocket and the molecule representation into a common feature space and scoring molecules directly based on their feature similarity to the target pocket. However, this approach suffers from overly simplistic negative pairs, which is inconsistent with the real-world virtual screening task. To solve this issue, we propose a self-supervised learning-based TTA approach to dynamically adjust the model to each test instance, aiming to enhance the performance in real virtual screening tasks.

**Test-Time Adaptation:** TTA is an emerging learning paradigm that adapts the model to target data at test time (Xiao & Snoek, 2024). TTA encompasses a variety of approaches (Xiao & Snoek, 2024), and the most widely used approach is based on auxiliary tasks, which have found extensive application in fields such as image processing (Yeo et al., 2023; Bahmani et al.; Park et al., 2024), video analysis (Lin et al., 2023; Xiong et al., 2024; Yi et al., 2023), 3D classification (Shim et al., 2025; Wang et al., 2024; Shin et al., 2022) and so on. Most studies utilizing this approach focus on how to design self-supervised learning tasks as auxiliary tasks and how to couple them with the primary task. For auxiliary task design, Varsavsky et al. (Varsavsky et al., 2020) introduce a framework combining adversarial loss with consistency regularization to enhance model adaptation at test time. CKEPE (Liu et al.) adapts the model by utilizing multimodal representation learning, and NC-TTT (Osowiechi et al., 2024) enhances self-adaptation by applying noise discrimination to the features. Besides designing proper auxiliary tasks, another important issue is how to couple them with the primary task to avoid overfitting the auxiliary tasks while harming the performance of the primary task. For example, Liu et al. (Liu et al., 2023) propose a novel meta-auxiliary learning framework aimed at optimizing model adaptation specifically for test-time scenarios. TT++ (Liu et al., 2021) proposes an online feature alignment strategy to align feature distributions at test time with those at training time, effectively mitigating overfitting to the auxiliary task. Point-TTA (Hatem et al., 2023) trains a meta-auxiliary learning TTA approach for point cloud registration. In this paper, we propose an adaptive algorithm for balancing self-supervised task losses, and utilize a meta-learning training framework to avoid bias towards auxiliary tasks.

## 3. Method

Given a protein pocket $p$ and a set of candidate molecules $M = \{m_1, m_2, \ldots, m_n\}$, the goal of virtual screening is to rank $M$ according to their probability of binding with $p$. In this work, we propose Drug-TTA, a test-time adaptation (TTA)-based virtual screening framework, which consists of a pocket encoder and molecule encoder, parameterized

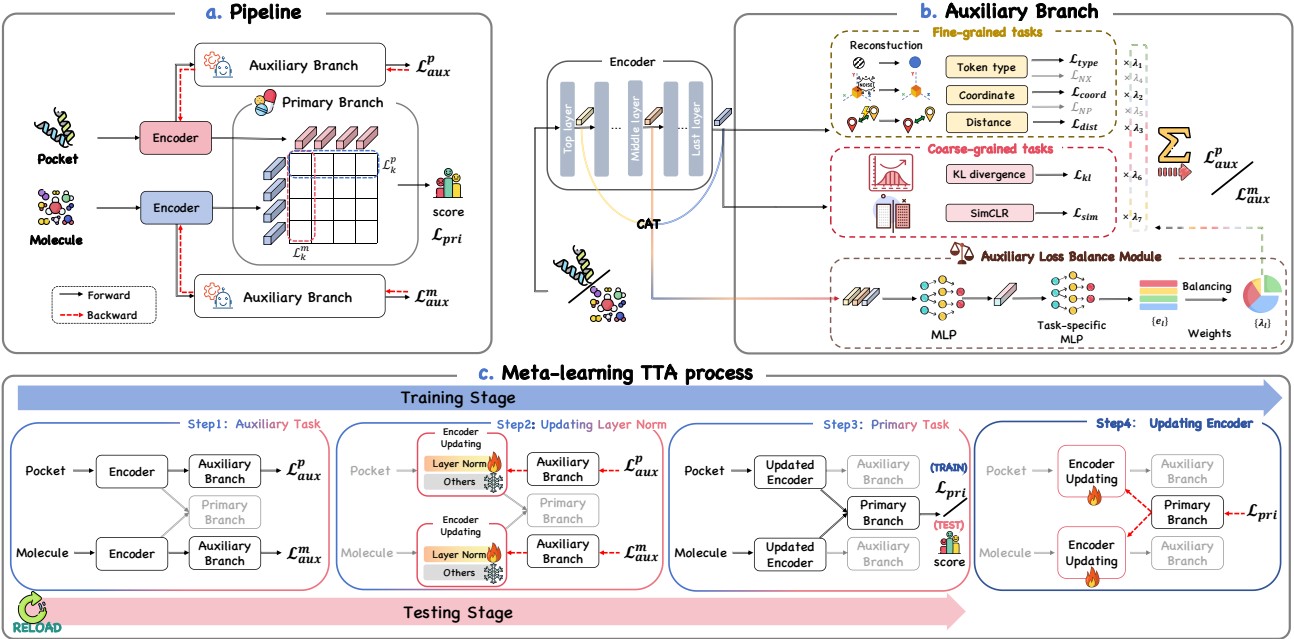

*Figure 2.* Overview of Drug-TTA. (a) Pipeline: The framework comprises two branches: a primary branch and an auxiliary branch. The input pockets and molecules are first encoded by their respective encoders. Then, the primary branch computes a similarity score between the encoded features of the pockets and the molecules to train the encoders through a contrastive loss. (b) Auxiliary branch: Pockets and molecules go through their own auxiliary branches, which have identical architecture but different parameters and inputs. Both auxiliary branches perform five self-supervised tasks, consisting of three fine-grained tasks and two coarse-grained tasks. Seven losses are calculated from the five auxiliary tasks and they are weighted and balanced through the multi-scale feature based Auxiliary Loss Balance Module (ALBM). This module concatenates features from the encoder's top, middle, and final layers, reduces their dimensionality through an MLP, and employs task-specific MLPs to generate adaptive weights for each loss. (c) Meta-learning TTA process: The training stage consists of four steps: (1) The pocket-ligand pairs are encoded through their respective encoders, and auxiliary task loss is computed. (2) The auxiliary loss is used to update the encoder's layer norm weights. (3) The updated encoder is employed to perform the primary task and calculate the primary loss. (4) All encoders are updated again by the primary loss. During the testing stage, the same pipeline is followed, and for each test instance, the encoder weights are first updated by the auxiliary loss, and then the final similarity score is calculated by the primary branch, which is used to rank the molecules.

by $\theta^p$ and $\theta^m$, respectively. The two encoders are trained to extract pocket and molecule features: $F_{\theta^p}(p) \rightarrow f_p$; $F_{\theta^m}(m) \rightarrow f_m$, and the molecules are ranked by the similarity between $f_p$ and $f_m$. Drug-TTA explores the TTA technique to enhance the model's generalization ability and overall performance. The model architecture and the training and testing workflow are illustrated in Figure 2, and it is further explained in the following three subsections.

In Section 3.1, we introduce the primary branch (Figure 2(a)), which learns feature alignment between pocket and molecule features through contrastive learning. Section 3.2 introduces the five auxiliary tasks and explains how we perform weight learning to balance them (Figure 2(b)). Finally, in Section 3.3, we describe the training and testing TTA process (Figure 2(c)).

### 3.1. Primary Branch

In this study, we follow the contrastive learning paradigm of DrugCLIP (Gao et al., 2024) to construct the virtual screening primary branch to perform the primary task. The

learning paradigm is briefly introduced below for completeness.

First, both protein pockets and molecules are represented at the atomic level, with atom types and 3D coordinates fed into the protein/molecule encoder. In this study, we follow the encoder architecture of Uni-Mol (Zhou et al., 2023). Through the encoder, both atom features and pairwise features are generated. Specifically, the atom features for pockets and molecules are encoded as $f_p$ and $f_m$, respectively. These representations are subsequently passed through simple non-linear projection heads $\sigma_{pri}^p$ and $\sigma_{pri}^m$, yielding the final output features $f_p'$ and $f_m'$.

During training, a contrastive learning approach is explored, where positive pairs are the input binding protein-molecule pairs. Similar to CLIP (Hafner et al., 2021), an in-batch sampling strategy is employed to generate negative pairs. Specifically, given a batch of $N$ input positive pairs $\{(x_k^p, x_k^m)\}_{k=1}^N$, with the list of pockets $\{(x_k^p)\}_{k=1}^N$ and the list of molecules $\{(x_k^m)\}_{k=1}^N$. The pockets and molecules can be combined to $N^2$ pairs $(x_i^p, x_j^m)$, where $i, j \in [1, N]$,

and pairs with $i = j$ are positive pairs, while those with $i \neq j$ serve as negative pairs.

The training loss consists of two parts: the Pocket-to-Mol loss $\mathcal{L}_k^p = -\frac{1}{N} \frac{exp(s(x_k^p, x_k^m)/\tau)}{\sum_i exp(s(x_k^p, x_i^m)/\tau)}$ and the Mol-to-Pocket loss $\mathcal{L}_k^m = -\frac{1}{N} \frac{exp(s(x_k^p, x_k^m)/\tau)}{\sum_i exp(s(x_i^p, x_k^m)/\tau)}$, where $s = \frac{f_p^T \cdot f_m}{\|f_p\|\|f_m\|}$ is the similarity functions composed of both the dot product($\cdot$) and cosine similarity, and $\tau$ represents the temperature parameter controlling the softmax distribution. The former measures the likelihood of ranking the binding molecule higher than other molecules for a given protein, while the latter evaluates the probability of correctly ranking the binding targets for a given molecule. By combining these two parts, the primary branch training loss is obtained as follows:

$$\mathcal{L}_{pri} = \frac{1}{2} \sum_{k=1}^{N} (\mathcal{L}_k^p + \mathcal{L}_k^m). \tag{1}$$

### 3.2. Auxiliary Branch

In this work, we introduce five auxiliary self-supervised tasks in the auxiliary branch to adapt the encoders to different data during inference. These tasks consist of three fine-grained tasks focusing on contextual information and two coarse-grained tasks focusing on modeling global features. The fine-grained tasks are masked atom type prediction, corrupted coordinate reconstruction and distance prediction, and the coarse-grained tasks are KL divergence minimization and SimCLR contrastive learning (Chen et al., 2020). These tasks generate seven distinct losses, with detailed descriptions provided in the Appendix A.

To better balance the aforementioned self-supervised tasks and their losses, one naive approach is to learn the weights for different losses during training and fix them for testing. However, this leads to overfitting of the weights to the training data, which contradicts the purpose of TTA—to adjust the model for better alignment with testing conditions. To this end, we propose the multi-scale feature based ALBM, which adjusts the weight allocation at test time based on the test instance's own feature.

As shown in Figure 2(b), a transformer architecture is used for both the protein and the molecule encoders, which consist of multiple layers. To capture a comprehensive representation of a test instance, we extract features from the top, middle and last layers of the encoder, which are denoted as $f_{top}$, $f_{mid}$ and $f_{last}$, respectively, and concatenate them. Subsequently, the concatenated features are passed through seven independent MLP heads $\{\text{MLP}_l\}$ $(l = 1, 2, .., 7)$, each responsible for calculating a weight for one of the seven losses. The weights $\{e_l\}$ $(l = 1, 2, ..., 7)$ are calculated as:

$$e_l = \text{MLP}_l \left(\text{Concat}(f_{top}, f_{mid}, f_{last})\right). \tag{2}$$

To prevent the model from collapsing during training and balance the distribution of weights, we apply a softmax function to constrain the weights, ensuring their sum equals one. As a result, final weights $\{\lambda_l\}$ $(l = 1, 2, .., 7)$ for the auxiliary task losses are mapped as follows:

$$\lambda_l = \text{Softmax} \left(\frac{1}{2e_l^2}\right). \tag{3}$$

The total auxiliary loss is formulated as:

$$\begin{aligned} \mathcal{L}_{aux} =& \lambda_1 \mathcal{L}_{type} + \lambda_2 \mathcal{L}_{coord} + \lambda_3 \mathcal{L}_{dist} \\ &+ \lambda_4 \mathcal{L}_{NX} + \lambda_5 \mathcal{L}_{NP} + \lambda_6 \mathcal{L}_{kl} + \lambda_7 \mathcal{L}_{sim}. \end{aligned} \tag{4}$$

### 3.3. Meta-learning TTA

Our goal is to leverage self-supervised auxiliary tasks to adjust the encoder weights during inference, enabling the model to quickly adapt its parameters for each test instance without the need for additional labels. To prevent the parameters updated by the auxiliary loss from becoming overly focused on improving the auxiliary tasks at the expense of the primary task, as suggested in (Hatem et al., 2023), we propose a meta-auxiliary task approach that jointly trains both the main and auxiliary tasks. There are three groups of parameters in the whole model, the pocket/molecule encoder parameters $\theta^p/\theta^m$, the pocket/molecule auxiliary branch head parameters $\sigma_{aux}^p/\sigma_{aux}^m$, and the pocket/molecule primary branch output head parameters $\sigma_{pri}^p/\sigma_{pri}^m$. We define three sets of parameters according to their different updating mechanism in training and inference. The updated parameter set in primary branch is $\theta_{pri} = \{\theta^p, \theta^m, \sigma_{pri}^p, \sigma_{pri}^m\}$, while the updated parameter sets in protein and molecule auxiliary branch are $\theta_{aux}^p = \{\phi_{norm}^p, \sigma_{aux}^p\}$ and $\theta_{aux}^m = \{\phi_{norm}^m, \sigma_{aux}^m\}$, where $\phi_{norm}^p$ and $\phi_{norm}^m$ are the parameters of normalization layers in $\theta^p$ and $\theta^m$.

As shown in Figure 2(c), the training and testing processes are largely similar, with the key difference occurring after the primary task is executed. During training, the encoder weights are updated based on the primary task loss by adjusting $\theta_{pri}$. In contrast, during testing, the model directly outputs the similarity score for the primary task without further updating the encoder weights.

**Training:** The initial weights for both encoders are obtained from pre-training on a large-scale dataset by Uni-Mol (Zhou et al., 2023). A batch of sample pairs $\{(x_k^p, x_k^m)\}_{k=1}^N$ is input, and the molecules and the pockets are separated and fed into their respective encoders and auxiliary task branches. The normalization layer parameters $\phi_{norm}^p$ and $\phi_{norm}^m$, as well as head parameters $\sigma_{aux}^p$ and $\sigma_{aux}^m$ are updated based on respective auxiliary task losses $\mathcal{L}_{aux}^p$ and $\mathcal{L}_{aux}^m$:

$$\begin{aligned} \{\phi_{norm}^p, \sigma_{aux}^p\} &\leftarrow \{\phi_{norm}^p, \sigma_{aux}^p\} - \alpha \nabla_\theta \mathcal{L}_{aux}^p \\ \{\phi_{norm}^m, \sigma_{aux}^m\} &\leftarrow \{\phi_{norm}^m, \sigma_{aux}^m\} - \alpha \nabla_\theta \mathcal{L}_{aux}^m, \end{aligned} \tag{5}$$

where $\alpha$ is the learning rate for auxiliary tasks.

As the primary branch has been updated by $\phi_{norm}^p$ and $\phi_{norm}^m$, the primary task loss is used to update all parameters of the primary branch:

$$\theta_{pri} \leftarrow \theta_{pri} - \beta \nabla_\theta \mathcal{L}_{pri}, \quad (6)$$

where $\beta$ is the learning rate for the primary task.

**Testing:** During testing, all trained model parameters are used. When a pocket $p$ and a set of candidate molecules $M$ are input, the normalization layer parameters as well as head parameters are first updated via the auxiliary tasks. Then, the updated encoders are directly used to perform the primary task. For the inference of a new iteration, the original parameters are reloaded to prevent accumulation offsets. Pseudo code for the training and the testing process are listed in Appendix C.

## 4. Experiments

To evaluate the performance of our method, we first assess the zero-shot performance of Drug-TTA on five virtual screening benchmarks: DUD-E (Mysinger et al., 2012), LIT-PCBA (Tran-Nguyen et al., 2020), AD (Chen et al., 2019), DEKOIS 2.0 (Bauer et al., 2013), and CASF-2016 (Su et al., 2018), comparing it with existing methods in Section 4.1. Here, "zero-shot" refers to the setting where the model is tested on entirely unseen protein pockets without using any labeled samples from the target benchmarks. Each virtual screening benchmark consists of multiple pockets, with each pocket corresponding to various active and inactive molecules (decoys). The task of virtual screening is to rank molecules such that active ones that bind to the corresponding pocket are placed at the top. To evaluate screening effectiveness, we use AUROC, BEDROC, and EF as our evaluation metrics. BEDROC is an improved version of AUROC, incorporating exponential weighting to emphasize to early rankings, while EF is a widely used virtual screening metric. Detailed definitions of evaluation metrics are in Appendix F. For training, we use the dataset from DrugCLIP (Gao et al., 2024), and when evaluating on each benchmark, we exclude it from the training datasets and retrain Drug-TTA. The hyperparameter settings for both training and testing are provided in Appendix E. Furthermore, we conduct ablation studies on key components, as detailed in Section 4.2. The visualization results can be found in Section 4.3.

### 4.1. Evaluation on virtual screening benchmarks

**DUD-E Benchmark** (Mysinger et al., 2012): DUD-E (Directory of Useful Decoys, Enhanced) Benchmark is a widely used virtual screening performance evaluation dataset. DUD-E contains 102 protein pockets and 22,886 active molecules, with an average of 224 active molecules

*Table 1.* Results on DUD-E. AUROC, BEDROC, and EF are reported (higher values indicate better performance). Bold values represent the best performance, and green indicates improvements of Drug-TTA over DrugCLIP.

| Method | AUROC(%) | BEDROC(%) | EF 0.50% | EF 1% | EF 5% |
|--------|----------|-----------|-------|-----|-----|
| Glide-SP | 76.70 | 40.70 | 19.39 | 16.18 | 7.23 |
| Vina | 71.60 | - | 9.13 | 7.32 | 4.44 |
| NN-score | 68.30 | 12.20 | 4.16 | 4.02 | 3.12 |
| RFscore | 65.21 | 12.41 | 4.90 | 4.52 | 2.98 |
| Pafnucy | 63.11 | 16.50 | 4.24 | 3.86 | 3.76 |
| OnionNet | 59.71 | 8.62 | 2.84 | 2.84 | 2.20 |
| Planet | 71.60 | - | 10.23 | 8.83 | 5.40 |
| DrugCLIP | 80.93 | 50.52 | 38.07 | 31.89 | 10.66 |
| Drug-TTA | **93.16** (↑12.23) | **82.82** (↑32.30) | **57.50** (↑19.43) | **54.04** (↑22.15) | **16.88** (↑6.22) |

per pocket. Each active molecule corresponds to 50 decoys, which are inactive molecules with similar physicochemical properties but different 2D topologies.

As shown in Table 1, we evaluate our method in a zero-shot setting and compare its performance with docking-based (Glide-SP (Halgren et al., 2004), Vina (Trott & Olson, 2010)), learning-based (NN-score (Durrant & McCammon, 2011), RFscore (Ballester & Mitchell, 2010), Pafnucy (Stepniewska-Dziubinska et al., 2018), OnionNet (Zheng et al., 2019), Planet (Zhang et al., 2023a), and feature alignment method (DrugCLIP (Gao et al., 2024)). The results clearly indicate that the feature alignment paradigm outperforms the other paradigms, showing that this paradigm is more suitable for virtual screening. Compared to Drug-CLIP, our method demonstrates a substantial performance improvement on all metrics, particularly in BEDROC, where we can observe an increase of 32.30%. This highlights that the TTA technique significantly enhances the model's generalization performance on unseen data.

**LIT-PCBA Benchmark** (Tran-Nguyen et al., 2020): LIT-PCBA (Large-scale Interaction and Toxicity Prediction for Chemical Bioactivity Assays) is a more challenging benchmark for virtual screening, designed to address the data bias issues found in other benchmarks such as DUD-E. It contains 15 pocket targets, 7,844 active molecules and 407,381 unique inactive molecules selected from high-confidence PubChem Bioassay data.

As shown in Table 2, we compare our method with docking-based methods (Surflex (Spitzer & Jain, 2012), Glide-SP (Halgren et al., 2004)), learning-based methods (Planet (Zhang et al., 2023a), Gnina (McNutt et al., 2021), Deep-DTA (Öztürk et al., 2018), BigBind (Brocidiacono et al., 2023)), SPRINT (McNutt et al., 2024)) and feature alignment method (DrugCLIP (Gao et al., 2024)). The results demonstrate that our method significantly improves the performance of DrugCLIP by a large margin, achieving SOTA performance across all metrics, with a particularly notable improvement in the EF metric with an order-of-magnitude enhancement.

*Table 2.* Results on LIT-PCBA. AUROC, BEDROC, and EF are reported (higher values indicate better performance). Bold values represent the best performance, and green indicates improvements of Drug-TTA over DrugCLIP.

| Method | AUROC(%) | BEDROC(%) | EF | | |
|---|---|---|---|---|---|
| | | | 0.50% | 1% | 5% |
| Surflex | 51.47 | - | - | 2.50 | - |
| Glide-SP | 53.15 | 4.00 | 3.17 | 3.41 | 2.01 |
| Planet | 57.31 | - | 4.64 | 3.87 | 2.43 |
| Gnina | 60.93 | 5.40 | - | 4.63 | - |
| DeepDTA | 56.27 | 2.53 | - | 1.47 | - |
| BigBind | 60.80 | - | - | 3.82 | - |
| SPRINT | **73.4** | 12.3 | 15.90 | 10.78 | 5.92 |
| DrugCLIP | 57.17 | 6.23 | 8.56 | 5.51 | 2.27 |
| Drug-TTA | 71.24 | **45.08** | **74.39** | **42.74** | **10.61** |
| | (↑14.07) | (↑38.85) | (↑65.83) | (↑37.23) | (↑8.34) |

*Table 3.* Results on AD, DEKOIS 2.0, and CASF-2016. AUROC, BEDROC, and EF are reported (higher values indicate better performance). Bold values represent the best performance, and green indicates improvements of Drug-TTA over DrugCLIP.

| Dataset | Method | AUROC(%) | BEDROC(%) | EF | | |
|---|---|---|---|---|---|---|
| | | | | 0.50% | 1% | 5% |
| AD | DrugCLIP | 81.19 | 52.04 | 20.50 | 18.00 | 9.10 |
| | Drug-TTA | **92.62** | **86.73** | **32.54** | **30.51** | **15.29** |
| | | (↑11.43) | (↑34.69) | (↑12.04) | (↑12.51) | (↑6.19) |
| DEKOIS 2.0 | DrugCLIP | 77.98 | 47.32 | 18.48 | 17.02 | 8.52 |
| | Drug-TTA | **83.61** | **73.64** | **26.41** | **25.64** | **13.06** |
| | | (↑5.63) | (↑26.32) | (↑7.93) | (↑8.62) | (↑4.54) |
| CASF-2016 | DrugCLIP | 85.92 | 67.88 | 36.19 | 34.19 | 12.93 |
| | Drug-TTA | **91.88** | **85.72** | **42.73** | **41.34** | **16.04** |
| | | (↑5.96) | (↑17.84) | (↑6.54) | (↑7.15) | (↑3.11) |

*Table 4.* Ablation Study on Various Aspects. "w/o" indicates the removal of the respective component compared to Drug-TTA.

| Ablation aspects | Method | AUROC(%) | BEDROC(%) | EF | | |
|---|---|---|---|---|---|---|
| | | | | 0.50% | 1% | 5% |
| - | DrugCLIP | 80.93 | 50.52 | 38.07 | 31.89 | 10.66 |
| | Drug-TTA | **93.16** | **82.82** | **57.50** | **54.04** | **16.88** |
| Weights Allocation | weights=1 | 83.63 | 51.75 | 39.15 | 33.05 | 11.29 |
| | fixed weight | 71.13 | 64.30 | 51.69 | 42.40 | 12.09 |
| | w/o weight regularization | 78.44 | 41.55 | 31.77 | 25.95 | 9.40 |
| Self-Supervised Task Selection | w/o coarse-grained tasks | 82.22 | 53.25 | 39.87 | 33.89 | 11.47 |
| | w/o fine-grained tasks | 79.35 | 41.77 | 31.58 | 26.05 | 9.53 |
| Auxiliary Branch Selection | w/o pocket TTA | 83.64 | 55.00 | 40.41 | 35.35 | 11.96 |
| | w/o molecule TTA | 79.85 | 41.55 | 31.24 | 26.08 | 9.49 |
| Multi-scale Feature Layer Selection | w/o middle and last layers | 83.55 | 58.12 | 43.54 | 36.95 | 12.47 |
| | w/o top and last layers | 84.38 | 59.78 | 45.42 | 38.31 | 12.56 |
| | w/o top and middle layers | 84.19 | 59.67 | 44.25 | 38.67 | 12.51 |
| | w/o last layer | 89.62 | 68.64 | 49.55 | 43.89 | 14.59 |
| | w/o top layer | 89.11 | 78.12 | 54.89 | 50.44 | 16.07 |
| | w/o middle layer | 82.60 | 67.09 | 50.98 | 43.92 | 13.47 |
| Meta-learning | w/o meta-learning | 80.19 | 40.90 | 31.19 | 25.14 | 9.46 |

molecules and 280 inactive molecules. Table 3 shows that Drug-TTA outperforms DrugCLIP across all metrics, demonstrating that TTA effectively enables the model to adapt to unseen datasets.

### 4.2. Ablation study

**Weights Allocation:** We conduct ablation experiments on the allocation of weights across different auxiliary task losses, and the results are shown in Table 4. First, we evaluate the model's performance without any weight allocation for the self-supervised tasks by setting all weights to 1 (the row "weights=1"), and the results indicate that the improvement over the baseline model (DrugCLIP) is minimal. Next, we evaluate the model's performance when using fixed weights after training, as shown in the row "fixed weight". Compared to DrugCLIP, we observe significant improvements across most metrics, except for AUROC, though the improvements are not as substantial as those seen with Drug-TTA. Furthermore, we assess whether using regularization constraints on the weights affects performance. The results in the row "w/o weight regularization" demonstrate that the absence of regularization leads to the failure of the TTA strategy, highlighting the importance of weight regularization.

**Self-Supervised Task Selection:** Given the large number of tasks used in Drug-TTA, we focus on ablation experiments involving the coarse-grained and fine-grained tasks. As shown in Table 4, using only fine-grained tasks ("w/o coarse-grained tasks") still results in performance improvements over DrugCLIP, although the gains are smaller. In contrast, when only coarse-grained tasks are used, the performance decreases compared to DrugCLIP. Significant performance gains are only achieved when both coarse-grained

**AD Benchmark** (Chen et al., 2019): This dataset is an improvement over DUD-E. Specifically, it reduces negative target bias by using active molecules from other targets as decoys for the current target. For each target, active molecules from 101 other targets are selected and docked in 101 separate batches, with the top 50 molecules by affinity chosen in each batch. This results in an average of 5,000 molecules used as decoys for each target. The comparison of our method with DrugCLIP is shown in Table 3. It can be observed that applying the TTA strategy significantly enhances the model's performance. When comparing the results in Table 1, we find that the improvements in the AD dataset over DUD-E have a minimal impact on the AUROC and BEDROC metrics for both DrugCLIP and Drug-TTA, but they significantly affect the EF metric.

**DEKOIS 2.0 Benchmark** (Bauer et al., 2013): DEKOIS 2.0 is a dataset specifically designed for evaluating virtual screening methods in drug discovery. It consists of 81 targets from different protein families, with each target having 40 active ligands and 1,200 decoys. As shown in Table 3, Drug-TTA outperforms DrugCLIP across all metrics, with a particularly notable improvement of 26.32% in BEDROC.

**CASF-2016 Benchmark** (Su et al., 2018): CASF-2016 Benchmark provides 57 targets for virtual screening tasks, with each target containing approximately 5 active

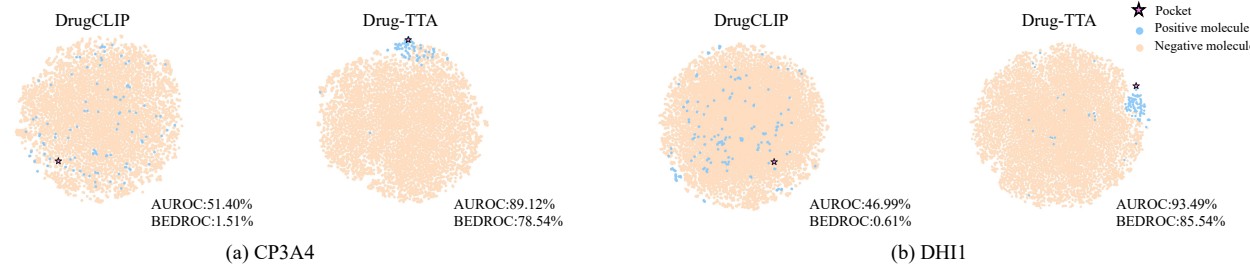

Figure 3. Visualization of feature distributions for the targets CP3A4 (a) and DHI1 (b) using DrugCLIP and Drug-TTA models. The red points represent pocket features, blue points denote active molecule features, and gray points indicate inactive molecule features.

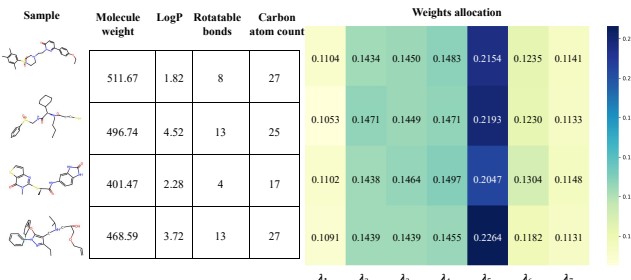

Figure 4. Weight allocation heatmap of four molecules in TTA and their molecular properties. The heatmap illustrates the weight allocation for four molecules during the TTA process. The corresponding molecular properties include molecular weight, LogP, the number of rotatable bonds, and carbon atom count, shown alongside the molecular structures.

and fine-grained tasks are combined (Drug-TTA), demonstrating the value of combining these task types for a more comprehensive learning of instance features. More details on the combination ablation of self-supervised tasks can be found in Appendix D.

**Auxiliary Branch Selection:** Drug-TTA applies TTA to fine-tune the encoders for both the pocket and molecule branches. As shown in Table 4, we evaluate the model's performance when TTA is applied to only one encoder—either the molecule encoder ("w/o pocket TTA") or the pocket encoder ("w/o molecule TTA"). The results show that excluding TTA from either encoder results in a performance decrease. Notably, applying TTA only to the pocket encoder results in worse performance than DrugCLIP, indicating that model adjustment requires simultaneous fine-tuning of paired data encoders to achieve optimal results.

**Multi-scale Feature Layer Selection:** In the process of generating loss weights controlled by instance features, our approach concatenates features from the encoder's top, middle, and last layers. We perform an ablation study on the selection of these layers, and the results are shown in Table 4. The results indicate that combining features from all three layers provides the best representation of the sample's features. Notably, the middle layer features demonstrate

greater importance than those from the top and last layers. For further insights of these three layers, additional t-SNE visualizations refer to Appendix G.

**Meta-learning:** To verify that meta-learning can effectively prevent the model from becoming biased toward the auxiliary task and producing suboptimal results, we conduct an ablation experiment on the meta-learning approach, with the results shown in Table 4. The results indicate that without the meta-learning based TTA approach (w/o meta-learning), the model's performance is even lower than DrugCLIP, suggesting that this approach leads to a performance decline in the primary task.

### 4.3. Visualization

We visualize the embeddings of two pocket targets and their corresponding molecules from the DUD-E dataset in Figure 3. The t-SNE plot illustrates both the pocket features and their corresponding active and inactive features. The visualization clearly demonstrates that, compared to Drug-CLIP, Drug-TTA significantly enhances the separability of active and inactive molecule features. Notably, the active molecule points are more tightly clustered around the pocket feature points, indicating that Drug-TTA improves feature alignment. Moreover, the performance metrics, including AUROC and BEDROC, show substantial improvements, with BEDROC increasing by an order of magnitude. This effect aligns with our objective of using feature alignment to identify molecules similar to a given pocket feature for virtual screening.

Furthermore, we analyze the weight allocation across different molecules. Figure 4 presents the heatmap of the weight distributions for four molecules with distinct chemical properties. This heatmap illustrates how the weight distribution varies according to different tasks, with the model adjusting the weight based on the inherent features of the test molecules. The varying weight assignments across molecules highlight the model's ability to dynamically allocate weight according to each sample, thereby allowing for more accurate focus on the most relevant features to improve performance.

# 5. Conclusion

In this paper, we introduce TTA for virtual screening for the first time. This approach effectively mitigates the issue of overly simplistic negative samples in the baseline model during training and better aligns with real-world virtual screening scenarios. Specifically, in such scenarios, both pockets and candidate molecules are unseen, and a large number of molecules fail to bind to the given pocket. Ablation study reveals that simply applying TTA to virtual screening is not sufficient. In our proposed framework, we incorporate abundant self-supervised auxiliary tasks and design an innovative module, multi-scale feature based ALBM, to generate weights for auxiliary task losses based on the features of each instance at test time, effectively addressing the task balancing issue. To further leverage the information from auxiliary tasks to improve the primary task, we adopt a meta-learning strategy to better couple the training and inference processes of both the primary task and the auxiliary tasks, preventing model bias towards the auxiliary tasks, which may negatively affect the primary task's performance. Significant performance improvement over current SOTA methods is achieved by the proposed method, which shows its great potential in virtual screening applications. In addition, this study shows the importance of TTA in virtual drug screening and this finding may spur many future work in this direction.

## Acknowledgments

This work is supported by the Science and Technology Innovation Plan Of Shanghai Science and Technology Commission grant 23S41900400, Fudan University Science Intelligence Special Fund grand FD-AI4S04183.

## Impact Statement

This paper presents work whose goal is to advance the field of Machine Learning. There are many potential societal consequences of our work, none which we feel must be specifically highlighted here.

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

## A. Auxiliary losses

In this work, we introduce five auxiliary self-supervised tasks in the auxiliary branch to adapt the encoders to different data during inference. These tasks consist of three fine-grained tasks focusing on contextual information and two coarse-grained tasks focusing on modeling global features. The fine-grained tasks are masked atom type prediction, corrupted coordinate reconstruction and distance prediction, and the coarse-grained tasks are KL divergence minimization and SimCLR contrastive learning (Chen et al., 2020).

**Masked atom type reconstruction:** This task aims to recover the atom type of masked atoms in a sample, mainly leveraging the local context provided by surrounding atoms. By doing so, it helps the encoder learn the relationships of chemical nature between atoms. The process begins by randomly masking a subset of atoms in the molecule or protein pocket. Some of the masked atoms are replaced with a special [MASK] token, while others are substituted with random tokens (Zhou et al., 2023). These masked molecules or pockets are then passed through their respective encoders to extract feature representations. Subsequently, the extracted features are fed into a prediction head to predict the original atomic types at the masked positions. The loss function is based on the negative log-likelihood of the predicted atomic types at the masked positions. Formally, the loss is expressed as:

$$\mathcal{L}_{type} = -\sum_{i=1}^{A} log(p(\hat{t}_i|t_i)), \tag{7}$$

where $\hat{t}_i$ is the predicted atom type for the $i$-th masked atom, $t_i$ is the ground-truth atom type, $p(\hat{t}_i|t_i)$ is the predicted probability, and $A$ is the total number of masked atoms. Following Uni-Mol (Zhou et al., 2023), an additional loss, $\mathcal{L}_{NX}$ is introduced to normalize masked atom features.

The following two reconstruction tasks follow this process, where the encoder output is used along with a prediction head for the reconstruction task, but with distinct output heads, each comprising a specific MLP.

**Corrupted coordinate reconstruction:** Building upon the previous task, this task is to reconstruct the 3D coordinates of the masked atoms, further enhancing the model's ability to understand spatial relationships and molecules' or pockets' geometry. Specifically, the same subset of masked atoms is considered, but their coordinates are now perturbed following the corrupted position generation and assignment algorithm of Uni-Mol (Zhou et al., 2023). The loss for this task is computed by comparing the predicted coordinates with the ground-truth coordinates and is defined as:

$$\mathcal{L}_{coord} = \frac{1}{A} \sum_{i=1}^{A} \mathcal{L}_{SmoothL1}(\hat{c}_i|c_i), \tag{8}$$

where $\hat{c}_i$ and $c_i$ are the predicted and ground-truth coordinate for the $i$-th masked atom, respectively.

**Distance reconstruction:** This task aims to predict the distances between pairs of masked atoms in the pockets or molecules, thus helping the model learn how atoms interact with each other in terms of their relative spatial positioning. For this task, all pairwise distances involving masked atoms from the previous task need to be reconstructed. The loss is calculated by comparing the predicted distances with the true ones and is defined as:

$$\mathcal{L}_{dist} = \frac{1}{B} \sum_{i=1}^{B} \mathcal{L}_{SmoothL1}(\hat{d}_i|d_i), \tag{9}$$

where $\hat{d}_i$ and $d_i$ are the predicted and ground-truth distance for the $i$-th pair to be reconstructed, and B is the total number of pairs that need to be reconstructed. To normalize masked pairwise features, an additional loss, $\mathcal{L}_{NP}$ is introduced following Uni-Mol (Zhou et al., 2023).

**KL divergence minimization:** This task encourages the model to learn a feature distribution that closely approximates a uniform distribution. As a result, the model is forced to learn more diverse feature representations, rather than excessively relying on specific patterns or features. Consequently, this regularization approach helps improve the model's generalization ability, enabling it to perform more effectively on unseen data. The KL loss $\mathcal{L}_{KL}$ is computed as the KL divergence between the softmaxed logits of latent features and a uniform distribution:

$$\mathcal{L}_{kl} = KL\left(softmax(Z)/\tau_{kl} \parallel U/\tau_{kl}\right), \tag{10}$$

where $Z = \{(f_k^p)\}_{k=1}^N$ or $\{(f_k^m)\}_{k=1}^N$ represents the latent features of the pockets or molecules, which are the output of their respective encoders, and $\tau_{kl}$ is a temperature scaling factor.

**SimCLR contrastive learning:** This contrastive learning task encourages the model to learn representations where similar pockets or molecules are mapped closer together in the feature space, while dissimilar pockets or molecules are pushed farther apart. This task enhances the model's ability to distinguish between different types of samples. Specifically, through the aforementioned reconstruction tasks, we obtain masked pockets $\{(\hat{x}_k^p)\}_{k=1}^N$ or molecules $\{(\hat{x}_k^m)\}_{k=1}^N$, which include both masked types and corrupted coordinates. These original and masked versions are treated as two distinct views of a pocket or a molecule. The features for each view are generated by their respective encoders, denoted as $z_k \in Z = \{(f_k^p)\}_{k=1}^N$ or $\{(f_k^m)\}_{k=1}^N$ and $\hat{z}_k \in \hat{Z} = \left\{\left(\hat{f}_k^p\right)\right\}_{k=1}^N$ or $\left\{\left(\hat{f}_k^m\right)\right\}_{k=1}^N$, respectively. SimCLR aims to minimize the distance between the original and masked versions of these two views $f_i^p$ and $\hat{f}_i^p$ ($f_i^m$ and $\hat{f}_i^m$) while maximizing the distance between them and all other views, $f_i^p$ and $\hat{f}_j^p$ ($f_i^m$ and $\hat{f}_j^m$) where $i \neq j$. The SimCLR loss can be formulated as follows:

$$
\begin{aligned}
\mathcal{L}_{\text{sim}} &= \mathbb{E}_{(z_i \sim Z, \hat{z}_j \sim \hat{Z})} \left[\ell_{z_i, \hat{z}_j}\right], \\
\ell_{z_i, \hat{z}_j} &= -\log \frac{\exp\left(\text{sim}\left(\ell_{z_i, \hat{z}_j}\right)/\tau_{\text{sim}}\right)}{\sum_{i \neq j} \exp\left(\text{sim}\left(\ell_{z_i, \hat{z}_j}\right)/\tau_{\text{sim}}\right)},
\end{aligned}
\tag{11}
$$

where $\text{sim}(\cdot)$ denotes the dot product and $\tau_{sim}$ is a temperature scaling factor.

## B. The hyperparameter settings

The hyperparameter settings for the main branch are exactly the same as those used in DrugCLIP(Gao et al., 2024), and the hyperparameter settings for the auxiliary branch model are shown in Table 5. To ensure that the losses from different tasks are on the same scale, we multiply each loss by a weight that controls the scale of each respective loss.

*Table 5.* Hyperparameter settings for auxiliary branch model.

| Hyperparameter | Value |
|---|---|
| $\tau_{kl}$ | 1.0 |
| $\tau_{sim}$ | 0.007 |
| Layers of weights allocation MLP | 2 |
| Layers of cat multi-layers of encoder MLP | 3 |
| Mask ratio | 0.15 |
| Corrupted distance | [-1,1] |
| Vocabulary size (atom types) | 30 (molecule) / 9 (pocket) |
| Gradient clip norm | 1.0 |
| Embedding dim of coarse-grained tasks | 128 |
| Embedding dim of fine-grained tasks | 512 |
| Control weight for | 1.0 (molecule) / 1.0 (pocket) |
| Control weight for $\mathcal{L}_{type}$ | 5 (molecule) / 1.0 (pocket) |
| Control weight for $\mathcal{L}_{coord}$ | 10 (molecule) / 1.0 (pocket) |
| Control weight for $\mathcal{L}_{dist}$ | 0.01 (molecule) / 0.01 (pocket) |
| Control weight for $\mathcal{L}_{NX}$ | 0.01 (molecule) / 0.01 (pocket) |
| Control weight for $\mathcal{L}_{NP}$ | 0.01 (molecule) / 0.01 (pocket) |
| Control weight for $\mathcal{L}_{kl}$ | 1000 (molecule) / 1000 (pocket) |
| Control weight for $\mathcal{L}_{sim}$ | 0.01 (molecule) / 0.01 (pocket) |

## C. Pseudo code

Pseudo codes for the training and the testing process are listed in Algorithm 1 and Algorithm 2, respectively.

---

**Algorithm 1** Training stage

---

**Require:** $\{(x_k^p, \ x_k^m)\}_{k=1}^N$: pairs of molecules and pockets; $\alpha, \beta$: learning rates

    Initialize the network with pre-trained weights $\theta^m, \theta^p$

    **for** batched pairs **do**

        Conduct auxiliary tasks in molecule branch: $\mathcal{L}_{aux}^m$

        Update parameters of auxiliary branch heads and encoder layer norm:

        $\{\phi_{norm}^m, \sigma_{aux}^m\} \leftarrow \{\phi_{norm}^m, \sigma_{aux}^m\} - \alpha\nabla_\theta\mathcal{L}_{aux}^m$

        Conduct the auxiliary tasks in pocket branch: $\mathcal{L}_{aux}^p$

        Update parameters of auxiliary branch heads and encoder layer norm:

        $\{\phi_{norm}^p, \sigma_{aux}^p\} \leftarrow \{\phi_{norm}^p, \sigma_{aux}^p\} - \alpha\nabla_\theta\mathcal{L}_{aux}^p$

        Update the primary task using the adapted parameters and update:

        $\theta_{pri} \leftarrow \theta_{pri} - \beta\nabla_\theta\mathcal{L}_{pri}$

    **end for**

    **Return** $\theta_{pri}, \theta_{aux}^m, \theta_{aux}^p$

---

**Algorithm 2** Testing stage

---

**Require:** An interesting pocket $p$ and a set of candidate molecules $M = \{m_1, m_2, \ldots, m_n\}$; $\eta$: learning rate

    Initialize the network with trained weights $\theta_{pri}$

    **for** batched molecules in $M$ **do**

        Conduct auxiliary tasks in molecule branch: $\mathcal{L}_{aux}^m$

        Update parameters of auxiliary branch heads and encoder layer norm:

        $\{\phi_{norm}^m, \sigma_{aux}^m\} \leftarrow \{\phi_{norm}^m, \sigma_{aux}^m\} - \eta\nabla_\theta\mathcal{L}_{aux}^m$

    **end for**

    Conduct the auxiliary tasks in pocket branch: $\mathcal{L}_{aux}^p$

    Update parameters of auxiliary branch heads and encoder layer norm:

    $\{\phi_{norm}^p, \sigma_{aux}^p\} \leftarrow \{\phi_{norm}^p, \sigma_{aux}^p\} - \eta\nabla_\theta\mathcal{L}_{aux}^p$

    Complete the primary task using the adapted parameters and compute scores:

    Scores $=$ Similarity$(F_{\theta^p}(p), F_{\theta^m}(M))$

    Reload $\theta_{pri}$ to prevent accumulation of offsets

    **Return** Predicted scores for each molecule

---

## D. Additional ablation study

We conduct ablation experiments on the selection of various auxiliary tasks, as shown in Table 6. $\mathcal{L}_{type}$ and $\mathcal{L}_{NX}$ are closely related, while $\mathcal{L}_{coord}$, $\mathcal{L}_{dist}$ and $\mathcal{L}_{NP}$ are also tightly linked. The ablation results reveal that without $\mathcal{L}_{type}$ and $\mathcal{L}_{NX}$, the model fails to converge, as demonstrated by the last two rows. The other results indicate that the inclusion of each loss term is meaningful.

Each row of the ablation study is explained in detail as follows:

- w/o $\mathcal{L}_{kl}$: Drug-TTA without the KL loss

- w/o $\mathcal{L}_{sim}$: Drug-TTA without the SimCLR loss

- w/o $\mathcal{L}_{type}$ $\mathcal{L}_{NX}$: Drug-TTA without the masked atom type construction loss and atom feature normalization loss

- w/o $\mathcal{L}_{coord}$ $\mathcal{L}_{dist}$ $\mathcal{L}_{NP}$: Drug-TTA without the corrupted coordinate reconstruction loss, distance reconstruction loss, and pairwise normalization loss

- w/o $\mathcal{L}_{coord}$ $\mathcal{L}_{dist}$ $\mathcal{L}_{NX}$ $\mathcal{L}_{NP}$: Drug-TTA without the corrupted coordinate reconstruction loss, distance reconstruction loss, atom feature normalization loss and pairwise feature normalization loss

- w/o $\mathcal{L}_{NX}$ $\mathcal{L}_{NP}$: Drug-TTA without the atom feature normalization loss and pairwise feature normalization loss

- w/o $\mathcal{L}_{type}$ $\mathcal{L}_{NX}$ $\mathcal{L}_{NP}$: Drug-TTA without the masked atom type construction loss, atom feature normalization loss and pairwise feature normalization loss

*Table 6.* Ablation study on self-supervised tasks.

| Method | AUROC(%) | BEDROC(%) | EF | | |
| --- | --- | --- | --- | --- | --- |
| | | | 0.50% | 1% | 5% |
| DrugCLIP(Gao et al., 2024) | 80.93 | 50.52 | 38.07 | 31.89 | 10.66 |
| Drug-TTA | **93.16** | **82.82** | **57.50** | **54.04** | **16.88** |
| w/o coarse-grained tasks | 82.22 | 53.25 | 39.87 | 33.89 | 11.47 |
| w/o fine-grained tasks | 79.35 | 41.77 | 31.58 | 26.05 | 9.53 |
| w/o $\mathcal{L}_{kl}$ | 88.38 | 69.72 | 50.30 | 44.61 | 14.60 |
| w/o $\mathcal{L}_{sim}$ | 83.71 | 54.76 | 41.59 | 35.09 | 11.60 |
| w/o $\mathcal{L}_{type}$ $\mathcal{L}_{NX}$ | 86.85 | 68.28 | 49.77 | 44.46 | 14.03 |
| w/o $\mathcal{L}_{coord}$ $\mathcal{L}_{dist}$ $\mathcal{L}_{NP}$ | 83.12 | 64.22 | 46.38 | 41.05 | 13.69 |
| w/o $\mathcal{L}_{coord}$ $\mathcal{L}_{dist}$ $\mathcal{L}_{NX}$ $\mathcal{L}_{NP}$ | 79.58 | 40.81 | 29.43 | 24.86 | 9.52 |
| w/o $\mathcal{L}_{NX}$ $\mathcal{L}_{NP}$ | - | - | - | - | - |
| w/o $\mathcal{L}_{type}$ $\mathcal{L}_{NX}$ $\mathcal{L}_{NP}$ | - | - | - | - | - |

Furthermore, we conduct an ablation study on the times of TTA fine-tuning for the encoder. We compare the performance of performing two adjustments to the molecule encoder before applying it to the primary task, as shown in Table 7. The results reveal that multiple adjustments yield inferior performance compared to a single update.

*Table 7.* Ablation study on TTA fine-tuning times.

| Method | AUROC(%) | BEDROC(%) | EF | | |
| --- | --- | --- | --- | --- | --- |
| | | | 0.50% | 1% | 5% |
| DrugCLIP(Gao et al., 2024) | 80.93 | 50.52 | 38.07 | 31.89 | 10.66 |
| Drug-TTA | **93.16** | **82.82** | **57.50** | **54.04** | **16.88** |
| TTA 2times | 84.04 | 71.76 | 54.97 | 47.13 | 14.03 |

# E. Details of training and testing

During both training and testing, we utilize the DrugCLIP dataset, which is derived from the PDBBind dataset (Wang et al., 2005) and augmented through the identification of homologous proteins. To achieve a zero-shot setup, we exclude any overlapping data from the benchmark dataset when evaluating on different datasets. In the training phase, we optimize the primary task using the AdamW optimizer with a learning rate of 1e-3 and a batch size of 48, with acceleration provided by an NVIDIA A40 GPU. For optimizing the auxiliary branch, we use the SGD optimizer, setting the learning rate for the molecule branch at 1e-3 and the pocket branch at 1e-4. During inference, we update only the auxiliary branch, continuing with the SGD optimizer; the learning rate for the molecule branch is set at 0.005, while the pocket branch's learning rate is 0.0001, and the batch size is increased to 64.

In the zero-shot inference process, DrugCLIP has not been evaluated on the AD, DEKOIS 2.0, and CASF-2016 benchmarks. The results on these three datasets are based on our own inference, while results on the other two datasets, including those of the comparison methods, are sourced from the original DrugCLIP paper. We reassess the overlap of data between the DrugCLIP training set and these three benchmarks. It is found that there is no data overlap between AD and CASF-2016, so we use the weights provided by DrugCLIP for inference on these two benchmarks. For DEKOIS 2.0, we remove the overlapping data and retrained the DrugCLIP model using its original hyperparameters.

## F. Evaluation metrics

**BEDROC (Boltzmann-Enhanced Discrimination of Receptor-Ligand Interactions):** BEDROC is a metric frequently applied in virtual screening, especially in drug discovery. It emphasizes the early-ranked molecules by incorporating an exponential weighting factor. The purpose of BEDROC is to evaluate the binding efficacy of molecules to the target, focusing primarily on the pocket-ligand interactions, while considering the relative importance of molecules ranked earlier in the screening process. $\text{BEDROC}_{85}$ is a common variant of BEDROC where the top 2% of candidates contribute 80% of the BEDROC score. The formula for BEDROC is defined as:

$$\text{BEDROC}_{\alpha} = \frac{\sum_{i=1}^{N} \exp\left(-\frac{\alpha r_i}{N}\right)}{Z_\alpha \left(\frac{1-\exp(-\alpha)}{\exp(\alpha/N)-1}\right)} \times \frac{Z_\alpha \sinh\left(\frac{\alpha}{2}\right)}{\cosh\left(\frac{\alpha}{2}\right) - \cosh\left(\frac{\alpha}{2} - \alpha Z_\alpha\right)} + \frac{1}{1 - \exp\left(\alpha(1 - Z_\alpha)\right)}, \tag{12}$$

where $\alpha$ is the parameter controlling the sensitivity of the metric to the early ranks in the list (typically ranges from 1 to 100), $N$ is the number of compounds being considered, $r_i$ is the rank of the $i$-th compound in the list, and $Z_\alpha$ is a normalization constant.

**EF (Enrichment Factor):** The Enrichment Factor (EF) is another commonly used metric for assessing how well active compounds are enriched within the top-ranked molecules from virtual screening. A higher EF suggests that the top-ranked molecules are more likely to be active. The EF is calculated as follows:

$$\text{EF}_{\alpha} = \frac{\text{NTB}_\alpha}{\alpha \cdot \text{NTB}_t}, \tag{13}$$

where $\text{NTB}_\alpha$ is the count of active molecules in the top $\alpha\%$ and $\text{NTB}_t$ is the total number of active compounds.

**ROC Enrichment Metric (RE):** The ROC enrichment metric (RE) is used to measure the ratio of the true positive rate to the false positive rate (FPR) at a specific threshold. It is defined as:

$$\text{RE}(x\%) = \frac{n \cdot \text{TP}}{\text{P} \cdot \text{FP}_{x\%}}, \tag{14}$$

where $n$ is the total number of compounds, TP refers to the true positive compounds correctly identified as active, P is the total number of active compounds, and $\text{FP}_{x\%}$ is the number of false positives predicted at a given threshold.

We add the comparison results between Drug-TTA and DrugCLIP on the RE metric across five tasks, as shown in Table 8. It can be observed that incorporating the TTA strategy significantly improves the performance.

*Table 8.* Additional results on RE metric.

| Dataset | Method | RE | | | |
|---|---|---|---|---|---|
| | | 0.50% | 1% | 2% | 5% |
| DUD-E | DrugCLIP | 73.97 | 41.79 | 23.68 | 11.16 |
| | Drug-TTA | **144.38** | **77.48** | **40.9** | **17.09** |
| LIT-PCBA | DrugCLIP | 9.03 | 5.54 | 3.46 | 2.27 |
| | Drug-TTA | **77.38** | **44.04** | **23.78** | **10.74** |
| AD | DrugCLIP | 52.44 | 33.07 | 20.63 | 10.57 |
| | Drug-TTA | **128.13** | **71.71** | **38.98** | **16.79** |
| DEKOIS 2.0 | DrugCLIP | 48.89 | 31.30 | 18.36 | 9.20 |
| | Drug-TTA | **107.84** | **58.18** | **31.68** | **13.73** |
| CASF-2016 | DrugCLIP | 96.33 | 53.09 | 30.42 | 13.43 |
| | Drug-TTA | **141.85** | **73.78** | **39.07** | **16.41** |

# G. Visualization

As shown in Figure 5, we visualize the features of active molecules (indicated in blue) and inactive molecules (indicated in grey) encoded by the top, middle, and last layers of the molecular encoder for two targets. We can observe that the features extracted by different layers represent different levels of molecular characteristics. Shallow-layer features, middle-layer features and deep-layer features exhibit significantly different distributions, suggesting that integrating features from all three layers is meaningful for generating loss weights.

Moreover, we visualize the feature distribution heatmap for additional molecules, as shown in Figure 6.

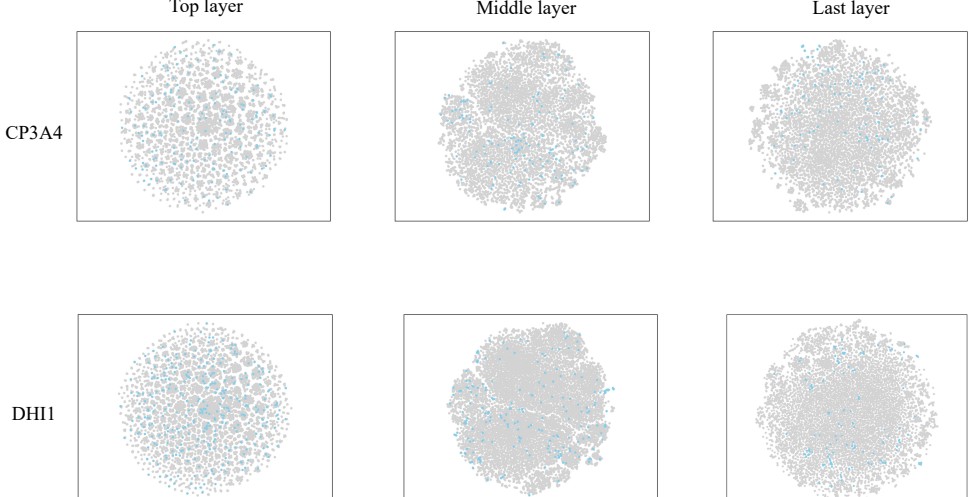

*Figure 5.* Visualization of feature distributions for the targets CP3A4 and DHI1 on the top, middle and last layer of molecule encoder. The blue points denote positive molecule features, and the gray points indicate negative molecule features.

| Sample | Molecule weight | LogP | Rotatable bonds | Carbon atom count | Weights allocation | | | | | | |
|---|---|---|---|---|---|---|---|---|---|---|---|
| | | | | | $\lambda_1$ | $\lambda_2$ | $\lambda_3$ | $\lambda_4$ | $\lambda_5$ | $\lambda_6$ | $\lambda_7$ |
| 1 | 512.65 | 2.94 | 16 | 29 | 1.0849 | 1.4734 | 1.4844 | 1.4900 | 2.0169 | 1.2921 | 1.1583 |
| 2 | 511.67 | 1.82 | 8 | 27 | 1.1041 | 1.4337 | 1.4497 | 1.4827 | 2.1537 | 1.2349 | 1.1412 |
| 3 | 473.60 | 1.77 | 7 | 22 | 1.0845 | 1.4749 | 1.4473 | 1.4693 | 2.1988 | 1.2080 | 1.1173 |
| 4 | 299.40 | -1.24 | 3 | 16 | 1.0932 | 1.4331 | 1.4490 | 1.5099 | 2.1039 | 1.2646 | 1.1464 |
| 5 | 492.32 | 3.36 | 8 | 17 | 1.0833 | 1.4733 | 1.4458 | 1.4297 | 2.2355 | 1.2028 | 1.1297 |
| 6 | 496.74 | 4.52 | 13 | 25 | 1.0527 | 1.4710 | 1.4491 | 1.4710 | 2.1932 | 1.2299 | 1.1331 |
| 7 | 449.62 | 4.76 | 6 | 26 | 1.0703 | 1.5011 | 1.4729 | 1.4674 | 2.0837 | 1.2692 | 1.1354 |
| 8 | 477.57 | -0.99 | 6 | 28 | 1.0836 | 1.4613 | 1.4237 | 1.4779 | 2.1979 | 1.2223 | 1.1333 |
| 9 | 485.99 | 3.24 | 8 | 26 | 1.0800 | 1.4386 | 1.4439 | 1.4765 | 2.1427 | 1.2438 | 1.1744 |
| 10 | 401.47 | 2.28 | 4 | 17 | 1.1016 | 1.4378 | 1.4643 | 1.4974 | 2.0470 | 1.3038 | 1.1481 |
| 11 | 407.49 | 1.17 | 5 | 18 | 1.0799 | 1.4176 | 1.5103 | 1.4654 | 2.0913 | 1.2685 | 1.1671 |
| 12 | 468.59 | 3.72 | 13 | 27 | 1.0908 | 1.4386 | 1.4386 | 1.4549 | 2.2644 | 1.1816 | 1.1310 |

*Figure 6.* Weight allocation heatmap of 12 molecules in TTA and their molecular properties. The heatmap illustrates the weight allocation for 12 molecules during the TTA process. The corresponding molecular properties include molecular weight, LogP, the number of rotatable bonds, and carbon atom count.

## H. Inference time and computational burden

We conduct additional experiments comparing the memory cost and inference time of Drug-TTA and DrugCLIP under the same conditions (i.e., on an RTX 3090 GPU with a batch size of 64).

**Memory Cost:** Compared to DrugCLIP, Drug-TTA increases memory usage from 1313 MiB to 1805 MiB for the molecule branch per batch and from 1232 MiB to 1627 MiB for the pocket branch (with a single target). Despite this increase, our model can still maintain a batch size of 64, ensuring training efficiency remains unaffected. Moreover, given that a standard 24 GB (24,576 MiB) GPU is widely accessible to researchers, this additional memory consumption remains well within practical limits and does not bring computational bottleneck.

**Inference Time:** Based on the average inference time under our experimental setup on the DUD-E benchmark, Drug-TTA requires 2.1 days for virtual screening at a practical scale (100 million molecules for a single target), compared to 0.8 day for DrugCLIP. While TTA does introduce additional computation time due to model adaptation during inference, this overhead is insignificant in the context of the overall drug discovery timeline, which spans years (including both in silico and wet-lab experiments). More importantly, the substantial performance improvement brought by Drug-TTA significantly reduces the trial-and-error burden in wet-lab experiments, making the additional computational time worthwhile.

In practice, given the relatively low hardware cost of our approach, inference time can be further reduced with minimal investment in computational resources if needed. We will incorporate this discussion on computational cost and inference time in the final version of the paper. Once again, we sincerely appreciate your interest in the practical applicability of our method.

