# OpenReview forum: "Drug-TTA: Test-Time Adaptation for Drug Virtual Screening via Multi-task Meta-Auxiliary Learning"
_ICML.cc/2025/Conference — ICML 2025 poster_

### Official Review · Reviewer_53jo · 2025-03-07

**Overall Recommendation:** 2

**Summary:**

The paper introduces Drug-TTA, a novel framework for drug virtual screening that incorporates test-time adaptation through multi-task meta-auxiliary learning. The authors build upon a contrastive learning paradigm by integrating a series of self-supervised auxiliary tasks-three fine-grained tasks (masked atom type prediction, corrupted coordinate reconstruction, and distance prediction), and two coarse-grained tasks (KL divergence minimization and SimCLR contrastive learning)-to dynamically adjust the model during inference. Central to their approach is the multi-scale feature-based Auxiliary Loss Balance Module (ALBM), which adaptively computes loss weights for each auxiliary task based on features extracted from different layers of the encoder. Moreover, a meta-learning strategy is employed to harmonize the auxiliary tasks with the primary virtual screening objective, ensuring that the test-time adaptations improve the main task without causing overfitting. Extensive experiments across five zero-shot virtual screening benchmarks demonstrate that Drug-TTA significantly outperforms existing methods, particularly showing notable improvements in AUROC, BEDROC, and enrichment factor metrics.

**Claims And Evidence:**

The claims are supported by experimental results showing improvements in AUROC, BEDROC, and enrichment factors across multiple benchmarks. However, the paper lacks efficiency analysis, which is crucial for assessing practicality, particularly in large-scale applications. Additionally, the generalization of the approach across diverse datasets and real-world scenarios is not sufficiently validated, weakening the overall claims.

**Essential References Not Discussed:**

None

**Experimental Designs Or Analyses:**

The experimental designs and analyses appear sound, with a clear comparison of Drug-TTA against baseline methods on multiple benchmark datasets. The metrics used, including AUROC, BEDROC, and enrichment factor, are standard and relevant for evaluating virtual screening methods.
However, there is no detailed analysis of the method's efficiency or scalability, which leaves questions about its practical deployment in large-scale settings unanswered.

**Methods And Evaluation Criteria:**

The proposed methods and evaluation criteria in the paper are appropriate for the problem at hand.
The evaluation criteria, including performance metrics such as AUROC, BEDROC, and enrichment factor, are standard and relevant for assessing the effectiveness of virtual screening methods.

**Other Comments Or Suggestions:**

None

**Other Strengths And Weaknesses:**

Strengths:
S1. The paper presents an innovative integration of test-time adaptation with multi-task meta-auxiliary learning, creatively combining contrastive learning with several self-supervised tasks. The proposed method addresses the domain shift between training and test data, which is a significant issue in virtual screening.
S2. It demonstrates significant performance improvements on multiple virtual screening benchmarks.
S3.The framework builds on recent advances (e.g., DrugCLIP, TTA) and addresses the challenging zero-shot setting in drug screening.

Weaknesses：
W1. The losses L_NX  and L_NP  appear in the framework diagram and loss function but are not explicitly introduced in the methodology. This omission makes it difficult for readers without prior knowledge to fully understand their role in the model.
W2. The ablation study table, table 6, does not clearly define what each experiment represents. For example, “w/o-L_kl” should be explicitly noted as “Drug-TTA without the KL loss” to improve clarity.
W3. The paper does not include any analysis of computational efficiency, scalability, or model complexity, which are crucial for assessing real-world applicability.
W4. The authors do not provide code or implementation details, making it difficult to verify whether the method can be reproduced in practice.

**Questions For Authors:**

Please address the concerns raised in W1–W4.

**Relation To Broader Scientific Literature:**

The Drug-TTA approach proposed in this paper echoes existing scientific literature in several aspects. First, the DrugCLIP framework significantly improves the performance of virtual screening by redefining virtual screening as a dense search task and using contrastive learning to align protein pockets and molecular representations.（https://arxiv.org/abs/2310.06367?utm_source=chatgpt.com）

In addition, the Point-TTA method adopts a test-time adaptation strategy in the point cloud registration task and designs three self-supervised auxiliary tasks to adapt to the new data distribution in the testing phase.（https://arxiv.org/abs/2308.16481?utm_source=chatgpt.com）

These studies provide theoretical foundations and practical experience for the Drug-TTA method, demonstrating the effectiveness of adaptive and contrastive learning strategies when applying tests in different tasks.

**Theoretical Claims:**

The paper does not include any formal proofs or rigorous theoretical claims that would require verification.

---

> ### Author Rebuttal · Authors · 2025-03-31
>
> Thank you for taking the time to review our paper and for your thoughtful comments. We sincerely appreciate your recognition of the novelty and significant performance of our method. Below, we provide detailed responses to your questions and concerns.
>
> ## W1: Losses $L_{NX}$ and $L_{NP}$ explanation
> We sincerely apologize for the insufficient explanation of $L_{NX}$ and $L_{NP}$. Below, we provide a more detailed clarification of these loss terms.
> Our encoder follows the overall framework of Uni-Mol, and the design of  $L_{NX}$ and $L_{NP}$ is aligned with its pretraining tasks. Specifically, $L_{NX}$ is introduced to normalize masked atom features. Given a molecular representation $X \in \mathbb{R}^{B \times T \times d}$, where $B$ is the batch size, $T$ is the number of atoms, and $d$ is the feature dimension, the norm constraint is formulated as:
>
> $
> \mathcal{L}_{NX} = \max(0, ||x||_2 - \sqrt{d} - \tau),
> $
> where $||x||_2$ ensures numerical stability, and $\tau$ is a tolerance margin. This constraint prevents masked atom features from deviating excessively from the expected norm, thereby stabilizing feature learning and preserving consistency across different masked regions.
>
> Similarly, $L_{NP}$ is applied to pairwise relational representations, ensuring that the learned attention patterns do not fluctuate excessively when certain atoms are masked. This is crucial because molecular graphs inherently depend on structural constraints, and unnormalized pairwise features may lead to unstable or biased attention distributions.
>
> By enforcing these norm constraints, our approach ensures that both atomic-level and relational representations remain well-calibrated, leading to more robust and generalizable molecular embeddings. We will revise the manuscript to incorporate this explanation and appreciate your insightful comments.
>
> ## W2: Table 6 revision
>
> Thank you very much for carefully reviewing our Appendix. We sincerely apologize for any lack of clarity in our original statement. Our revised version is as follows:
>
> - Drug-TTA without coarse-grained tasks
> - Drug-TTA without fine-grained tasks
> - Drug-TTA without the KL loss
> - Drug-TTA without the SimCLR loss
> - Drug-TTA without the masked atom type construction loss and atom feature normalization loss
> - Drug-TTA without the corrupted coordinate reconstruction loss, distance reconstruction loss, and pairwise normalization loss
> - Drug-TTA without the corrupted coordinate reconstruction loss, distance reconstruction loss, atom feature normalization loss and pairwise feature normalization loss
> - Drug-TTA without the atom feature normalization loss and pairwise feature normalization loss
> - Drug-TTA without the masked atom type construction loss, atom feature normalization loss and pairwise feature normalization loss
>
> ## W3: Concern regarding computational efficiency, scalability, or model complexity
> Thank you for raising this question. We apologize for not providing a detailed explanation in the paper. To address this issue, we conduct additional experiments comparing the memory cost and inference time of Drug-TTA and DrugCLIP under the same conditions (i.e., on an RTX 3090 GPU with a batch size of 64).
> - **Memory Cost**: Compared to DrugCLIP, Drug-TTA increases memory usage from 1313 MiB to 1805 MiB for the molecule branch per batch and from 1232 MiB to 1627 MiB for the pocket branch (with a single target). Despite this increase, our model can still maintain a batch size of 64, ensuring training efficiency remains unaffected. Moreover, given that a standard 24 GB (24,576 MiB) GPU is widely accessible to researchers, this additional memory consumption remains well within practical limits and does not bring computational bottleneck.
> - **Inference Time**: Based on the average inference time under our experimental setup on the DUD-E benchmark, Drug-TTA requires 2.1 days for virtual screening at a practical scale (100 million molecules for a single target), compared to 0.8 days for DrugCLIP. While TTA does introduce additional computation time due to model adaptation during inference, this overhead is insignificant in the context of the overall drug discovery timeline, which spans years (including both in silico and wet-lab experiments).  More importantly, the substantial performance improvement brought by Drug-TTA significantly reduces the trial-and-error burden in wet-lab experiments, making the additional computational time worthwhile.
>
> In practice, given the relatively low hardware cost of our approach, inference time can be further reduced with minimal investment in computational resources if needed.  We will incorporate this discussion on computational cost and inference time in the final version of the paper. Once again, we sincerely appreciate your interest in the practical applicability of our method.
> ## W4: Code release
> We assure you that upon acceptance, we will release our original code and model weights to facilitate reproducibility.

---

### Official Review · Reviewer_doCT · 2025-03-13

**Overall Recommendation:** 3

**Summary:**

The authors pinpoint two issues in ML-based structure-based drug discovery: a lack of negative pairs for ML-based docking methods and overly simplistic negative pairs in contrastive learning approaches (e.g., DrugClip), leading to a domain shift during inference when most screened molecules are inactive.
To mitigate this, they propose a test-time adaptation pipeline centered on DrugClip that updates the encoders using multiple auxiliary tasks, where the weight of each task is adjusted per input, and using meta-learning to prevent overfitting to these auxiliary tasks.
The authors show that their methods, Drug-TTA, significantly improved over their baseline method DrugCLIP and is the new SOTA.

**Claims And Evidence:**

Evidence for the claims is clear and convincing.

**Essential References Not Discussed:**

No

**Experimental Designs Or Analyses:**

- In general, the experimental design is sound using and their analysis seems sound.
- Furthermore, the proposed methods is quite complex with many components and the authors have done a systematic ablation study on each of the components.
- However, not a single value in the paper comes with an error bar giving limited significance to the results.
   I'm very aware this is systematic problem with the community, and these error bars are never reported on these VS benchmarks, but you can do better.
  I'm not asking you to produce triplicates for all of your experiments, but I would like to see replicates and the analysis for at least one of the datasets (e.g., CASF-2016).

**Methods And Evaluation Criteria:**

The proposed method was evaluated on established virtual screening benchmarks using standard metrics, and compared against both well-established and previous state-of-the-art approaches.
However, the evaluation does not include a comparison to the recent SPRINT [McNutt'24] method, which is, to my knowledge, the current SOTA for LIT-PCBA.

[McNutt'24] SPRINT Enables Interpretable and Ultra-Fast Virtual Screening against Thousands of Proteomes, http://arxiv.org/abs/2411.15418

**Other Comments Or Suggestions:**

- C1: In figure 3, finding the pocket representation and the contrast between the actives and inactives is not very clear.
   Maybe increase the size of pocket representation dot and change the color scheme.
- C2: In the introduction maybe clarify that your are talking about *structure-based* virtual screening.
- C3: Values in the Fig 4. heatmap are way too small.
   Also is 4 decimals really required in a illustration?

**Other Strengths And Weaknesses:**

- S1: Paper is generally well-written.
- S3: Figure 1 very nice and gives an excellent overview of the method!
- W1: With virtual screening libraries like GDB-17 and Enamine Real now containing billions of compounds, the efficiency of VS methods is more critical than ever.
   While Drug-TTA shows promise, it introduces additional complexity over DrugClip by incorporating extra parameters and performing parameter updates during inference.
   A direct comparison of parameter counts and inference times between DrugClip and Drug-TTA would be invaluable for assessing the practical efficiency of the proposed method.

**Questions For Authors:**

- Q1: As I understand the text and equation in 3.2, the concatenated features directly are passed to the seven different MLPs, but in Figure 1 it seems that the concatenated features are first passed in to a common MLP before having a separate MLP for each task.
   Can you clarify on this?

**Relation To Broader Scientific Literature:**

Building on prior work in virtual screening (VS), test-time adaptation (TTA), and meta-learning, the proposed approach, Drug-TTA, enhances generalizability by: (1) introducing TTA to drug virtual screening, and (2) achieving state-of-the-art performance on several VS benchmarks.

**Theoretical Claims:**

Application paper, doesn't contain proof.
In general, didn't check the correctness of the formulas.

---

> ### Author Rebuttal · Authors · 2025-03-31
>
> Thank you for taking the time to review our paper and for your thoughtful and thorough feedback. We truly appreciate your recognition that the evidence for the claims is clear and convincing, as well as your positive comments on our writing and figures. Below, we will address your questions and concerns in detail.
> ## Methods and Evaluation Criteria: additional comparison method
> Thank you for the suggestion. We will discuss this method and cite the new arXiv paper in the final version. While our method does not surpass SPRINT [1] in AUROC on LIT-PCBA, it achieves improvements in BEDROC and EF. We will incorporate this comparison accordingly.
> Performance Comparison:
> |Method| AUROC(%)| BEDROC(%)|EF 0.50% |EF 1%|EF 5%|
> |-|-|-|-|-|-|
> |DrugCLIP|57.17|6.23| 8.56 | 5.51| 2.27|
> |Drug-TTA|71.24| **45.08**| **74.39**|**42.74**|**10.61**|
> |SPRINT[1]| **73.4**|12.3|15.90|10.78|5.92|
>
> [1] [McNutt'24] SPRINT Enables Interpretable and Ultra-Fast Virtual Screening against Thousands of Proteomes, http://arxiv.org/abs/2411.15418
> ## Experimental Designs or Analyses: replicates on all datasets
> Thank you very much for raising this question. We conduct additional testing of our method on five benchmarks, and the results are averaged over three repetitions. Below are the mean and standard deviation for each benchmark:
> |Benchmark| AUROC(%)| BEDROC(%)| EF 0.50%| EF 1%| EF 5%|
> |-|-|-|-|-|-|
> |DUD-E|93.15±0.01|82.72±0.27|57.47±0.09|53.90±0.13|16.85±0.06|
> |LIT-PCBA|71.19±0.19|44.87±0.06|74.26±0.24|42.44±0.37|10.43±0.03|
> |AD|92.68±0.08|86.80±0.10|32.56±0.03| 30.56±0.06|15.31±0.03|
> |DEKOIS2.0|83.43±0.24|73.30±0.31|26.63±0.28|25.43±0.21|12.94±0.11|
> |CASF-2016|91.82±0.08|86.01±0.25|42.84±0.12|41.49±0.19|16.05±0.02|
>
> ## W1: Concern regarding inference time and computational burden
> Thank you for raising this question. We apologize for not providing a detailed explanation in the paper. To address this issue, we conduct additional experiments comparing the memory cost and inference time of Drug-TTA and DrugCLIP under the same conditions (i.e., on an RTX 3090 GPU with a batch size of 64).
> - **Memory Cost**: Compared to DrugCLIP, Drug-TTA increases memory usage from 1313 MiB to 1805 MiB for the molecule branch per batch and from 1232 MiB to 1627 MiB for the pocket branch (with a single target). Despite this increase, our model can still maintain a batch size of 64, ensuring training efficiency remains unaffected. Moreover, given that a standard 24 GB (24,576 MiB) GPU is widely accessible to researchers, this additional memory consumption remains well within practical limits and does not bring computational bottleneck.
> - **Inference Time**: Based on the average inference time under our experimental setup on the DUD-E benchmark, Drug-TTA requires 2.1 days for virtual screening at a practical scale (100 million molecules for a single target), compared to 0.8 day for DrugCLIP. While TTA does introduce additional computation time due to model adaptation during inference, this overhead is insignificant in the context of the overall drug discovery timeline, which spans years (including both in silico and wet-lab experiments). More importantly, the substantial performance improvement brought by Drug-TTA significantly reduces the trial-and-error burden in wet-lab experiments, making the additional computational time worthwhile.
>
> In practice, given the relatively low hardware cost of our approach, inference time can be further reduced with minimal investment in computational resources if needed. We will incorporate this discussion on computational cost and inference time in the final version of the paper. Once again, we sincerely appreciate your interest in the practical applicability of our method.
> ## C1: Redrawing Figure 3
> Thank you for your suggestion regarding our visualizations. We redraw Figure 3 with an increased size of the pocket representation, improved color schemes, and clearer markers for better readability. Since images cannot be uploaded here, please refer to the updated figure at https://pasteboard.co/Up1mGfvCKsYN.bmp.
> ## C2: Clarification on structure-based virtual screening
> Thank you for your suggestion. We will clarify that we are talking about *structure-based* virtual screening in the Introduction.
> ## C3: Value size and decimal explanation
> We appreciate your feedback. We redraw Figure 4 and enlarge the values in the middle heatmap, as shown in https://pasteboard.co/chAPEluE4BNN.bmp. The four-decimal precision is chosen to better show the differences in sample weights.
> ## Q1: Clarification of framework diagram
> Thank you for your thorough review. We sincerely apologize for any misunderstanding caused by the design of our main framework diagram. We will clarify this in the paper. The common MLP is just employed to project the concatenated three-layer features (512 × 3 = 1536) into 128 dimensions. The subsequent seven MLPs are specifically designed to accommodate seven auxiliary losses. We will update our description in the Method.

---

### Official Review · Reviewer_neHb · 2025-03-13

**Overall Recommendation:** 4

**Summary:**

The paper introduces Drug-TTA, a novel test-time adaptation (TTA) framework for drug virtual screening that dynamically adjusts a pre-trained model to each test instance. Unlike traditional zero-shot screening methods, Drug-TTA utilizes self-supervised auxiliary tasks to adapt its protein and molecule encoders at inference time, enhancing generalization to unseen data. A key innovation is the multi-scale feature-based Auxiliary Loss Balance Module (ALBM), which optimally balances auxiliary task losses. Additionally, meta-learning ensures that adaptation enhances rather than hinders the primary task. Drug-TTA significantly outperforms state-of-the-art (SOTA) methods across five benchmarks (DUD-E, LIT-PCBA, AD, DEKOIS 2.0, CASF-2016), with an average AUROC improvement of 9.86%, demonstrating its effectiveness in zero-shot-like virtual screening while not being strictly zero-shot due to test-time updates.

**Claims And Evidence:**

Yes

**Essential References Not Discussed:**

Related works are properly discussed.

**Experimental Designs Or Analyses:**

Yes, but there seems to be a difference in evaluation settings with comparison baselines. The baseline models are zero-shot settings, but strictly speaking, Drug-TTA does not seem to be a zero-shot setting (due to Algorithm 2).

**Methods And Evaluation Criteria:**

Yes

**Other Comments Or Suggestions:**

The table captions are poor and need to be improved.

**Other Strengths And Weaknesses:**

[Strengths]

- Incorporating five self-supervised auxiliary tasks (both fine-grained and coarse-grained) allows the model to learn useful representations without labeled binding affinity data, making it adaptable and efficient.

- Drug-TTA outperforms existing methods across five major virtual screening benchmarks, achieving an average AUROC improvement of 9.86% over DrugCLIP, demonstrating its superior accuracy in ranking active molecules.

[Weaknesses]

- While it operates in a zero-shot setting, Drug-TTA modifies model parameters during test-time inference, which contradicts the strict definition of zero-shot learning, making it less interpretable in direct zero-shot comparisons.

- Test-time adaptation requires multiple forward and backward passes per test instance, increasing inference time and computational burden compared to traditional zero-shot methods like DrugCLIP.

- The paper reports a significant improvement in performance on the LIT-PCBA benchmark, particularly in EF (Enrichment Factor) metrics, where Drug-TTA achieves an order-of-magnitude improvement over DrugCLIP. However, the paper does not provide a thorough discussion on why Drug-TTA performs exceptionally well on LIT-PCBA compared to other benchmarks.

**Questions For Authors:**

Please see the weaknesses.

**Relation To Broader Scientific Literature:**

In the case of molecules, it is difficult to find good negative samples, and we want to address this through test-time adaptation. This seems to have high utility not only in virtual screening but also in other molecular property prediction fields.

**Theoretical Claims:**

Not available.

---

> ### Author Rebuttal · Authors · 2025-03-31
>
> Thank you for taking the time to review our paper and for your thoughtful feedback. We greatly appreciate your recognition of our innovation and the practical significance of our work. Below, we provide detailed responses to your concerns.
> ## W1: Concern regarding "Zero-Shot" strict definition
> We sincerely appreciate your comment. You are correct that our method involves updating model parameters during testing, which does not align with the definition of zero-shot learning. Nevertheless, as you commented, our method does work in a zero-shot-like setting. Because of the very similar working condition, some recent TTA [1-3] works also define methods like ours, where no labeled samples are used during testing, as zero-shot. We will clarify the difference in revision.
>
> We acknowledge that our comparison methods do not adjust model parameters at test time, which may raise concerns about fairness. However, our input conditions remain identical across all methods, as we do not use any labeled test samples. Moreover, since our work is the first to introduce TTA in this task, there are no directly comparable baselines. Given these constraints, we compare against the most relevant existing approaches. We appreciate your insightful remark and will refine the description of zero-shot learning in the final version of our paper.
> ### References:
> [1] Liberatori B, Conti A, Rota P, et al. Test-time zero-shot temporal action localization. CVPR 2024.
> [2] Zhao S, Wang X, Zhu L, et al. Test-Time Adaptation with CLIP Reward for Zero-Shot Generalization in Vision-Language Models. ICLR 2024.
> [3] Aleem S, Wang F, Maniparambil M, et al. Test-time adaptation with SALIP: A cascade of SAM and CLIP for zero-shot medical image segmentation. CVPR 2024.
> ## W2: Concern regarding inference time and computational burden
> We appreciate your concern regarding inference time and computational burden of our method. Due to response length constraints, we kindly refer you to our response to Reviewer doCT(W1), where we provide a detailed analysis.
> ## W3: Performance analysis on LIT-PCBA
> Thank you for carefully reviewing our experimental results and for your valuable suggestions. We supplement our analysis with additional experimental and visualization results as follows.
>
> LIT-PCBA is a highly challenging task due to its extreme class imbalance, with only 0.74% of the screened molecules being active. This makes all methods have much lower performance on this dataset than on the other benchmarks and leaves much room for improvement. Drug-TTA, with its ability to adapt to each sample, is particularly well-suited to handling such challenging scenarios.
>
> Specifically, in the figure of https://pasteboard.co/Go21VXS5RV5H.bmp, we visualize the feature distributions for the MTORC1 target on the LIT-PCBA benchmark using both Drug-TTA and DrugCLIP. The visualization clearly shows that positive molecules are positioned closer to the pocket features in Drug-TTA, resulting in higher ranking scores for these molecules. Consequently, this enhances the early retrieval capability, which explains the substantial improvements in BEDROC and EF—metrics that are highly sensitive to early ranking performance.
> Furthermore, we summarize the performance improvements observed for several targets:
> | Target  | Active Molecule Proportion | AUROC (Drug-TTA) | BEDROC (Drug-TTA) | AUROC (DrugCLIP) | BEDROC (DrugCLIP) |
> |-|-|-|-|-|-|
> | **MTORC1** | 0.29% | 98.81% | 48.38% | 67.38% | 1.74% |
> | **PKM2**   | 0.22% | 93.71% | 59.31% | 74.28% | 0.90% |
> | **FEN1**   | 0.10% | 98.35% | 58.36% | 87.51% | 1.79% |
>
> From the table, it is evident that Drug-TTA is more robust to extreme class imbalance. This adaptation significantly enhances early retrieval performance.
>
> We must **clarify that** our performance on LIT-PCBA only surpasses the performance on DUD-E in one metric, EF (0.50%). This is because EF focuses solely on the top-ranked molecules, and given the low proportion of active molecules in the dataset, EF can be more easily improved as active molecules become more concentrated at the top. However, regarding AUROC and BEDROC, which are influenced by the overall ranking, the performance on LIT-PCBA is lower compared to other benchmarks.
>
> ## Other Comments or Suggestions: Table captions need improvement
> Thank you for your comment on the table caption. Since we are unable to modify the original manuscript at this stage, we will make the revisions in the final version of the paper. Specifically, we plan to add the following clarifications: In Table 123: "AUROC, BEDROC, and EF are reported (higher values indicate better performance). Bold values represent the best performance, and green indicates improvements of Drug-TTA over DrugCLIP." In Table 4: "*"w/o"* indicates the removal of the respective component compared to Drug-TTA."

---

### Official Review · Reviewer_wTg7 · 2025-03-14

**Overall Recommendation:** 2

**Summary:**

This paper introduces Drug-TTA, a novel test-time adaptation (TTA) approach for drug virtual screening that leverages multi-task meta-auxiliary learning to adapt the model to each test instance. Drug-TTA incorporates a large number of self-supervised auxiliary tasks into both training and inference processes and proposes an Auxiliary Loss Balance Module (ALBM) based on multi-scale features to dynamically adjust the weights of auxiliary tasks during inference. Extensive experiments demonstrate its effectiveness.

**Claims And Evidence:**

Yes

**Essential References Not Discussed:**

NA

**Experimental Designs Or Analyses:**

Yes

**Methods And Evaluation Criteria:**

Yes

**Other Comments Or Suggestions:**

NA

**Other Strengths And Weaknesses:**

Strengths:

1. The proposed method significantly outperforms the baseline DrugCLIP on multiple datasets.
2. The experimental section is comprehensive.
3. The writing is clear, particularly with the well-illustrated framework diagram.


Weakness:

1. Parameter fine-tuning is required during the TTA (Test-Time Adaptation) phase, and new targets all need to be re-fine-tuned, which may incur additional computational overhead.
2. I'm a bit puzzled by the analysis of Figure 5. Could the authors provide a more detailed analysis and explanation?
3. It seems that there are numerous hyperparameters to design and many tricks involved. Although the authors conducted ablation studies on almost every module, I found that the performance drops significantly when one setting is missing, which is somewhat puzzling. On one hand, this indicates that the workload is substantial. However, on the other hand, it may increase the difficulty of reproducing the results.
4. As is well known, the Lit-PCBA dataset is relatively challenging. Why does the proposed method achieve such significant performance improvements on the Lit-PCBA dataset compared to other baselines, and even surpass the performance on simpler tasks like DUD-E? I believe more analysis is needed to explain why the method works so well in this context.

**Questions For Authors:**

I have a few detailed questions:


1. How did you consider adjusting the normalization layers in the encoder using auxiliary tasks during test time? What would happen if all parameters were fine-tuned?
2. How long does it take to fine-tune during the test phase, and is it necessary for the loss of the auxiliary task to converge?

**Relation To Broader Scientific Literature:**

Refer to the strengths and weakness.

**Theoretical Claims:**

This paper does not include much theoretical part.

---

> ### Author Rebuttal · Authors · 2025-03-31
>
> Thank you for taking the time to review our paper and for your thoughtful feedback. We greatly appreciate your recognition of the novelty and effectiveness of our work, as well as your positive remarks on our writing and presentation. In response to your comments, we conduct additional experiments and provide further visualization analysis. Below are our point-by-point responses to your concerns.
> ## W1&Q2 (part 1): Concern regarding additional computational overhead and inference time
> We appreciate your concern regarding computational overhead and inference time of Drug-TTA. Due to response length constraints, we kindly refer you to our response to Reviewer doCT(W1), where we provide a detailed analysis.
> ## W2: More detailed analysis and explanation of Figure 5
> Thank you for carefully reviewing our appendix. As shown in Figure 2(b), our ALBM concatenates features from the molecule encoder's top, middle, and last layers and uses the concatenated features to calculate weights for each loss to balance the auxiliary losses. Figure 5 illustrates the feature space distribution of molecules at different encoder layers (top, middle, and last) during virtual screening across different targets. Our visualization highlights the diversity in molecular feature representations at different layers, demonstrating the necessity of fusing features from the three layers in ALBM. This observation is also consistent with the results of our ablation study in Table 4, which demonstrates that fusing features from all three layers is essential for optimal performance.
> ## W3: More detailed analysis of ablation study results
> Thank you for acknowledging the substantial workload of our ablation studies. We understand your concern regarding the significant performance drop when removing certain components. This is because the key components of Drug-TTA are highly interdependent, with each component playing a crucial role. Below, we provide a detailed explanation:
> - **Auxiliary Branch Selection**: Performing TTA on only one branch (either molecule or pocket) leads to a mismatch in feature adaptation, as the unadapted branch struggles to align within the shared feature space. Thus, TTA is necessary for both the molecule and pocket branches.
> - **Multi-Scale Feature Layer Selection**: Our framework is designed to comprehensively capture molecular features across multiple layers. While removing any single layer leads to a performance drop, the model still outperforms DrugCLIP, demonstrating the robustness of our multi-scale feature design.
> - **Self-Supervised Task Selection**: Self-supervised tasks are crucial for learning both fine-grained and coarse-grained molecular representations, where coarse-grained features aggregate fine-grained representations. Removing the fine-grained feature learning task results in suboptimal performance improvement.
> - **Weight Regularization & Meta-Learning**: Weight regularization prevents model collapse during training. Meta-learning is the core mechanism that prevents overfitting to auxiliary tasks. Removing these components leads to a significant performance drop, highlighting their necessity.
>
> Overall, most ablation settings still outperform DrugCLIP, validating the effectiveness of our design. We assure you that upon acceptance, we will release our original code and model weights to facilitate reproducibility.
> ## W4: Performance analysis on LIT-PCBA
> We appreciate your concern regarding the performance analysis on LIT-PCBA. Due to response length constraints, we kindly refer you to our response to Reviewer neHb(W3), where we provide a detailed analysis.
> ## Q1: How did you consider adjusting the normalization layers in the encoder using auxiliary tasks during test time? What would happen if all parameters were fine-tuned?
> Thank you for your question. We conduct an additional experiment with full fine-tuning, and the results on the DUD-E dataset are shown below. Fine-tuning all parameters with limited data leads to instability due to the large parameter space and distribution shift, causing gradient explosion or model collapse. This ultimately degrades performance, as reflected in our results. Therefore, we consider adjusting the normalization layers, as in other TTA methods.
> |Method|AUROC(%)|BEDROC(%)|EF 0.50%|EF 1%|EF 5%|
> |-|-|-|-|-|-|
> |**DrugCLIP**|80.93|50.52|38.07|31.89|10.66|
> |**Drug-TTA**|93.16|82.82|57.50|54.04|16.88|
> |**All parameters fine-tuned**|64.34|11.32|7.20|5.97|3.81|
> ## Q2(Part 2): Is it necessary for the loss of auxiliary task to converge?
> Thank you for your question. Our method does not require the auxiliary task loss to converge, as the model only needs to perform a single adjustment during testing. This aligns with the common practice in TTA, where most methods apply only one or a few updates rather than waiting for full convergence. This design ensures efficiency and generalizability, preventing excessive computation and overfitting to individual test samples.

---

### Decision · Program_Chairs · 2025-05-01

**Decision:**

Accept (poster)

**Comment:**

The paper introduces Drug-TTA, a novel test-time adaptation (TTA) method for drug virtual screening. The approach leverages self-supervised auxiliary tasks and meta-learning to dynamically adapt the model to each test instance, aiming to improve the performance of virtual screening tasks.


Based on the reviews and rebuttal discussions:
- The paper presents a novel, well-executed methodology with significant performance gains.
- Key concerns about computational burden and hyperparameter fine-tuning were acknowledged but sufficiently addressed.
- While there remain concerns about efficiency, the authors demonstrated clear advantages over existing methods.

Given these considerations, it is a boarderline paper. The recommendation leans toward acceptance, if there is room in the program.